# MIST: Moment-Aligned Invariant Stability Transform for Robust Flow Matching

Liang Peng [* 1]   Deqing Li [* 1]   Yujia Wu [1]   Hao Meng [1]   Kuan Cao [1]   Yu Wu [1]   Xiaoxiao Xu [1]   Lin Qu [1]

## Abstract

Classifier-Free Guidance (CFG) is a cornerstone of flow-matching models, significantly enhancing visual quality and prompt adherence. However, high guidance scales inherently violate the optimal transport dynamics, leading to visual artifacts and mode collapse. In this paper, we investigate the mechanisms of this failure through the lens of velocity moment decomposition. Our analysis reveals that the distributional shift induced by CFG decouples into two geometric components: a **Linear Barycentric Drift** that shifts the global distribution center, and a **Quadratic Energetic Instability** that injects surplus kinetic energy, disrupting the transport cost and triggering variance explosion. To mitigate these issues, we introduce **MIST** (**M**oment-aligned **I**nvariant **S**tability **T**ransform), a training-free method designed to confine the sampling trajectory to the learned data manifold. MIST comprises two hierarchical stages: (1) **Invariant Alignment (IA)**, a global statistical rectifier that restores structural integrity by removing the linear drift and realigning the energy profile; and (2) **Stability Thresholding (ST)**, a local dynamical regulator that enforces Lipschitz-like smoothness via temporal decay and spatial suppression. MIST enables robust, high-fidelity generation across a wide range of guidance scales while consistently improving performance at moderate scales. Extensive experiments on diverse text-to-image and text-to-video benchmarks demonstrate that MIST outperforms standard CFG and state-of-the-art corrections, establishing a new benchmark for robust guidance in flow-based generative models.

---

[*]Equal contribution   [1]Alibaba Group, Hangzhou, China. Correspondence to: Liang Peng <jingjie.pl@alibaba-inc.com>.

*Proceedings of the 43 $^{rd}$ International Conference on Machine Learning*, Seoul, South Korea. PMLR 306, 2026. Copyright 2026 by the author(s).

## 1. Introduction

Flow matching models (Lipman et al., 2022; Esser et al., 2024) have emerged as a leading paradigm in generative modeling, setting new standards in image and video synthesis. Their success stems not only from architectural innovations but also from effective guidance methods that steer generation toward user intent. Among these, Classifier-Free Guidance (CFG) (Ho & Salimans, 2022) is widely used in improving visual fidelity and prompt alignment.

CFG amplifies the influence of conditioning signals during the iterative denoising process via a single hyperparameter: the guidance scale $w$. Intuitively, higher values of $w$ should yield stronger prompt adherence. However, in practice, this introduces a fundamental trade-off: increasing $w$ beyond moderate levels inevitably triggers severe instabilities, manifesting as visual artifacts, color over-saturation, and structural collapse. These issues severely constrain the robustness of flow models, preventing users from leveraging high guidance scales to achieve maximal prompt alignment.

This work investigates the underlying causes of CFG instability. We reveal that this guidance induces a distributional shift in the predicted velocity field. By analyzing the transport dynamics, we identify that this shift decouples into two detrimental components: a **linear barycentric drift** that pushes the global distribution center away from the learned manifold, and a **quadratic energetic instability** that injects excessive kinetic energy into the sampling trajectory. As illustrated in Figure 1, at $w = 15$, these unconstrained forces cause standard CFG to generate over-saturated and stylistically biased images, signaling a catastrophic collapse of the underlying statistical distribution.

To address these challenges, we propose **MIST** (**M**oment-aligned **I**nvariant **S**tability **T**ransform), a hierarchical framework designed to stabilize the sampling process by enforcing constraints derived from our moment analysis. MIST comprises two core components: **Invariant Alignment (IA):** A global statistical rectifier that targets the distributional shift. By neutralizing the *Linear Barycentric Drift* (first moment) and renormalizing the *Quadratic Energetic Instability* (second moment), IA ensures that the guided velocity remains statistically consistent with the learned manifold, ef-

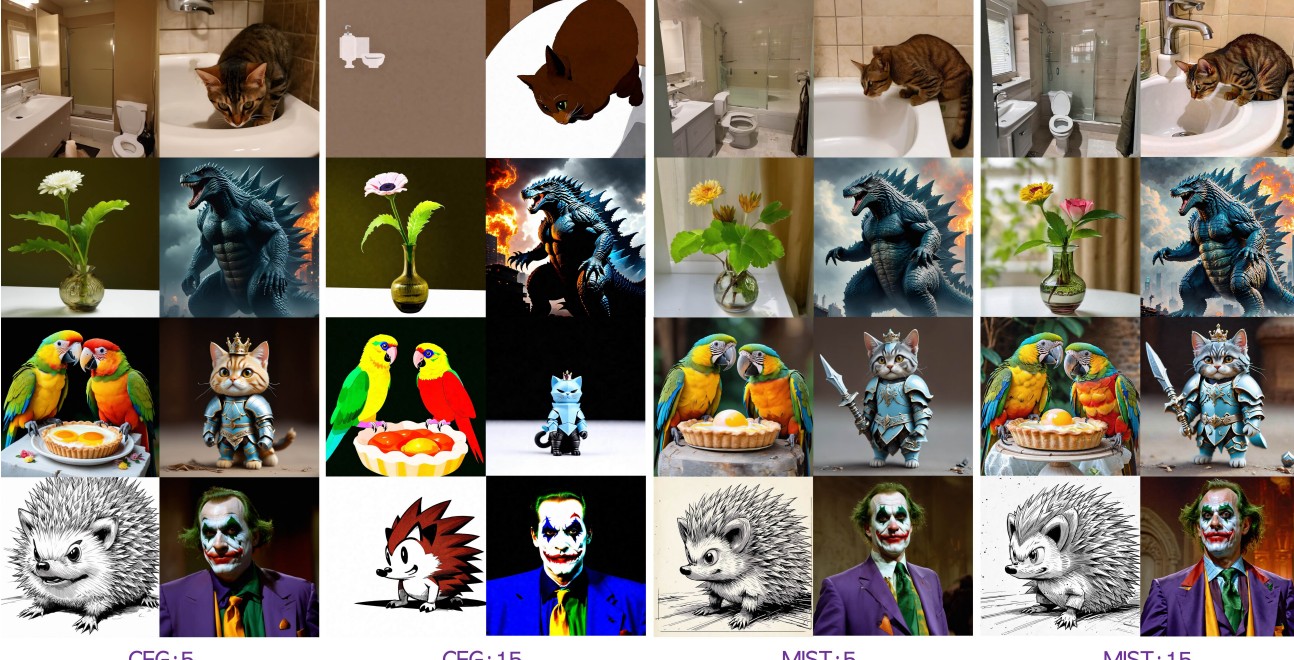

CFG:5        CFG:15        MIST:5        MIST:15

*Figure 1.* Comparisons between CFG and MIST at different guidance scales. MIST offers two benefits. First, it avoids mode collapse at high scales ($w = 15$, CFG tends to generate overly simplified and stylized (*e.g.*, anime-like) images); Second, it provides more visually appealing results on moderate scales ($w = 5$). Results are generated using SD3.5 (Esser et al., 2024) with the same random seed.

fectively preventing mode collapse (see Figure 2). **Stability Thresholding (ST):** A local dynamical regulator that tames the quadratic term. It employs a temporal constraint across the denoising process to ensure trajectory stability, while imposing spatial regularization to suppress local numerical singularities caused by excessive guidance forces.

Crucially, MIST confers a dual advantage over existing methods: it significantly extends the operational range of guidance, enabling high-fidelity generation even at extreme scales where standard CFG collapses, while simultaneously enhancing visual detail and prompt adherence at moderate scales (see Figure 1; full prompts are in the Appendix). Our contributions can be summarized as follows:

- We identify the detrimental velocity distribution shift as the fundamental cause of high-CFG instability, formally decomposing it into geometrically distinct linear drift and quadratic energy components.

- We propose **MIST**, a training-free and plug-and-play method for flow-based models, combining **Invariant Alignment (IA)** and **Stability Thresholding (ST)** to enforce moment constraints and dynamical stability.

- We validate MIST across state-of-the-art flow models, including SD3, Flux-dev, and Wan2.2. Experiments demonstrate that our approach consistently outperforms standard CFG and recent variants, establishing a new state-of-the-art for guidance in both image and video generation tasks.

## 2. Related Work

### 2.1. Flow Matching Diffusion Models

Diffusion models have set a new benchmark for high-fidelity image and video synthesis. Early advances (Song & Ermon, 2019; Song et al., 2020b; Sohl-Dickstein et al., 2015; Nichol et al., 2021; Blattmann et al., 2023) are predominantly SDE-based, with methods such as DDPM (Ho et al., 2020), DDIM (Song et al., 2020a), EDM (Karras et al., 2022; 2024), Stable Diffusion (Rombach et al., 2022; Podell et al., 2023; Lin et al., 2024), and DiT (Peebles & Xie, 2023) modeling stochastic diffusion dynamics via SDEs. More recently, flow-based approaches grounded in flow matching (Lipman et al., 2022) have emerged as the mainstream: they formulate generation as a deterministic ODE by learning a time-dependent velocity field that transports samples from a simple prior to the data distribution, leading to more stable training and improved interpretability. Motivated by this perspective, a series of text-to-image models, including Rectified Flow (Liu et al., 2022), SD3/SD3.5 (Esser et al., 2024), Lumina-Next (Zhuo et al., 2024), and Flux (Labs, 2024; Labs et al., 2025), as well as text-to-video models (Guo et al., 2023; Ma et al., 2025; Team, 2024; HaCohen et al., 2024) such as HunyuanVideo (Kong et al., 2024) and Wan2.1/2.2 (Wan et al., 2025) employ velocity-based training and sampling. Accordingly, our study centers on flow-based models as the primary vehicle for analysis and method design.

## 2.2. Classifier-free Guidance for Diffusion Models

Aligning text prompts with image and video generations remains a central yet challenging problem. Early methods used classifier guidance (Dhariwal & Nichol, 2021), injecting gradients from an external classifier. This approach induces training and compatibility overhead. Classifier-free guidance (CFG) (Ho & Salimans, 2022) removes the external classifier by jointly training conditional and unconditional models and blending their predictions at inference via a tunable guidance scale. However, this scale is an empirical hyperparameter whose mis-specification can cause artifacts or under-conditioning. To address these issues, some works (Zheng & Lan, 2023; Xia et al., 2025; Wang et al., 2024; Yehezkel et al., 2025) introduce adaptive or time-varying schedules to improve the guiding process. Some other works (Sadat et al., 2023) focus on enhancing the diversity of generations. Other approaches like (Kynkäänniemi et al., 2024) limit guidance to specific sampling intervals. Further refinements to CFG include APG (Sadat et al., 2025), which decomposes the CFG update term into parallel and orthogonal components and removes the parallel component to reduce oversaturation. CFG++ (Chung et al., 2025) reformulates text-guidance as an inverse problem with a text-conditioned score matching loss, thereby tackling the off-manifold challenges inherent in traditional CFG. More recently, to improve flow-based models, CFG-Zero (Fan et al., 2025) optimizes the scale by velocity projections and proposes zero-initialization for the first few steps. In summary, the evolution of text-guided generation techniques highlights a continuous effort to achieve more precise, efficient, and robust alignment between textual prompts and visual outputs. While progress has been made, most methods still struggle with stability at high guidance scales. Our work complements these efforts by unifying global statistical alignment and local dynamical stabilization under a single moment-guided framework.

## 3. Theoretical Analysis of Guidance-Induced Distribution Shift

In flow-matching models, the generation process is governed by a time-dependent velocity field $v_t$. Under Classifier-Free Guidance (CFG), the effective velocity is constructed as an affine transformation: $v_t^{\text{CFG}} = v_t(x) + w \cdot \delta v_t$, where $v_t(x)$ denotes the unconditional velocity and $\delta v_t$ represents the difference between the conditional and unconditional velocities. While this effectively enhances prompt adherence, high guidance scales $w$ introduce severe instability.

To formalize this instability, we analyze the impact of $w$ on the transport dynamics. Specifically, the distributional shift induced by CFG can be characterized by examining its effects on the first two statistical moments. Let $\mathcal{D}_{\text{shift}}(p_t^w)$ denote the perturbation of the transport dynamics induced

by the guidance scale $w$. Conceptually, this term captures how the evolution of the probability density $p_t$ is distorted by the additional guidance term. We characterize it through its projection onto the velocity moments. This allows us to decouple the instability into two geometrically distinct components:

$$
\mathcal{D}_{\text{shift}}(p_t^w) \sim \begin{cases} \underbrace{w \cdot \mathbb{E}_{x \sim p_t}[\delta v_t]}_{\text{Linear Barycentric Drift (1st Moment)}} \\ \underbrace{\frac{1}{2} w^2 \cdot \mathbb{E}_{x \sim p_t}\left[\delta v_t^\top \mathcal{M}(x)\, \delta v_t\right]}_{\text{Quadratic Energetic Instability (2nd Moment)}} \end{cases} \quad (1)
$$

Here, $\mathcal{M}(x)$ represents the local metric of the data manifold. From an information-geometric perspective, this aligns with the Fisher Information Matrix (Amari & Nagaoka, 2000), which quantifies how sensitive the generated distribution is to local perturbations. While we typically assume a simplified Euclidean geometry ($\mathcal{M} \approx \mathbf{I}$), the underlying semantic manifold remains inherently curved. Retaining $\mathcal{M}(x)$ allows us to capture this non-uniformity: regions with high information density (i.e., large eigenvalues of $\mathcal{M}$), such as complex textures or object boundaries, act as "amplifiers" for the quadratic energy term. $\mathcal{M}(x)$ is currently motivational rather than operational, which provides an information-geometric interpretation of why certain regions may amplify instability. Consequently, even a moderate guidance scale $w$ can trigger excessive instability in these sensitive areas. This decomposition reveals the dual nature of failure modes at high $w$:

- **The Linear Term:** The first-order term, scaling linearly with $w$, represents the *Barycentric Drift*. Since $\mathbb{E}[v_t^{\text{CFG}}] = \mathbb{E}[v_t] + w\mathbb{E}[\delta v_t]$, this term describes a rigid translation of the probability mass's center. Intuitively, this pushes the global distribution away from the learned manifold, leading to misalignment.
- **The Quadratic Term:** The second-order term, scaling with $w^2$, represents the *Quadratic Energetic Instability*. In the optimal transport framework for flow matching, the trajectory's smoothness is governed by the transport cost, defined as the kinetic energy $\mathcal{E} \propto \|v\|^2$ (Benamou & Brenier, 2000). CFG artificially injects a surplus energy term proportional to $w^2\|\delta v_t\|^2$. This massive injection of kinetic energy violates the minimal-cost principle of the learned prior, manifesting as an uncontrolled explosion of variance. **Locally**, in regions with high gradient conflict or high curvature (large $\mathcal{M}(x)$), this energy leads to sharp numerical singularities and trajectory oscillations.

## 4. Methodology: The MIST Method

Motivated by the above decomposition, we propose the **MIST** (**M**oment-aligned **I**nvariant **S**tability **T**ransform), a hierarchical intervention strategy. **IA (Invariant Alignment)** first restores global statistical integrity by neutralizing

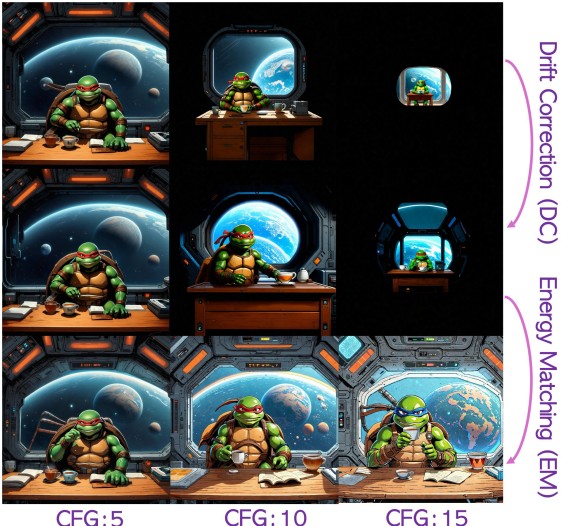

CFG:5     CFG:10     CFG:15

*Figure 2.* Standard CFG suffers from mode collapse at high scales ($w = 15$), Applying Drift Corrections (DC) alone alleviate this problem. Combining with Energy Matching (EM) fully restores the statistical nature, yielding a balanced image with correct lighting and geometry. Best viewed in color with zoomed in.

the first-order linear drift and realigning the second-order variance scale. Complementarily, **ST (Stability Thresholding)** acts as a local dynamical regulator to suppress numerical singularities and ensure the stability of the sampling trajectory. Together, they enforce a rigorous manifold constraint on the guidance process. Unlike higher-order moments which affect distribution shape, these first two moments represent the sufficient statistics governing the trajectory's stability and integrity.

### 4.1. Invariant Alignment (IA)

The **Invariant Alignment** operator acts as a *global statistical rectifier*. It targets the distributional shift identified in our analysis by explicitly correcting the first two moments of the velocity field.

**Drift Correction (DC).** Our analysis identifies the $O(w)$ term as a linear drift that shifts the probability mass. To remedy this, we project the guidance vector $\delta v_t$ onto the zero-mean subspace:

$$\delta v_t^{dc} = \delta v_t - \mathbb{E}_{x \sim p_t}[\delta v_t]. \quad (2)$$

By enforcing $\mathbb{E}[\delta v_t^{dc}] = 0$, we eliminate the global shift. This anchors the expectation of the guided velocity to that of the unconditional prior, i.e., $\mathbb{E}[v_t^{dc}] = \mathbb{E}[v_t(x)]$, preventing the global misalignment.

**Energy Matching (EM).** The more severe instability arises from the $O(w^2)$ quadratic term, which triggers a kinetic energy explosion. To address this, we impose an energy constraint, treating the variance of the unconditional velocity as the target equilibrium. We realign the velocity:

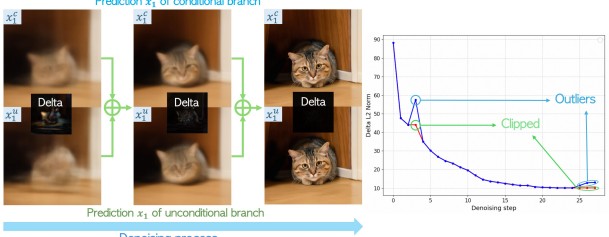

*Figure 3.* Temporal Decay (TD). The guidance magnitude exhibits sharp spikes during the sampling process, causing trajectory oscillation. TD imposes a monotonicity constraint, effectively damping these sudden bursts. This enforces temporal smoothness (Lipschitz continuity) on the ODE trajectory, ensuring stable convergence from noise to data. Prompt: *A cat in a house.*

$$v_t^{IA} = \mu + (v_t^{dc} - \mu) \cdot \frac{\sigma(v_t(x))}{\sigma(v_t^{dc}) + \epsilon}, \quad (3)$$

where $\mu = \mathbb{E}[v_t(x)]$, $\sigma(\cdot)$ denotes the spatial standard deviation, and $\epsilon = 1e^{-8}$ ensures numerical stability. This operator effectively renormalizes the kinetic energy, ensuring the trajectory remains within the valid energy profile of the learned manifold. This prevents the variance collapse and artifacts. We provide an example in Figure 2.

### 4.2. Stability Thresholding (ST)

While IA restores global statistical integrity, local regions with high curvature (large $\mathcal{M}(x)$) remain prone to the numerical singularities caused by the quadratic term. We introduce Stability Thresholding as a local dynamical regulator.

**Temporal Decay (TD).** We model the generation process as a dynamical system. From a numerical analysis perspective, theoretical convergence requires the velocity field to maintain temporal smoothness (Lipschitz continuity) (Ascher & Petzold, 1998). However, high guidance scales often induce energy spikes, causing the ODE to become stiff and unstable. To mitigate this, we impose a monotonic regularity constraint. To accommodate the natural volatility of early structural formation, we skip the initial $T_{clip}$ steps and apply:

$$\hat{\delta v}_t = \delta v_t \cdot \min\left(1, \frac{\|\delta v_{t+1}\|_2}{\|\delta v_t\|_2}\right). \quad (4)$$

This effectively clips gradients that violate the energy decay principle ($\dot{\mathcal{E}} \leq 0$), suppressing oscillatory divergence. Conceptually, this enforces an annealing schedule (Song & Ermon, 2019), ensuring a smooth, convergent transition from the chaotic high-energy noise distribution to the structured data manifold. We illustrate TD in Figure 3.

**Spatial Suppression (SS).** Visual artifacts can arise locally when the guidance force overwhelms the learned unconditional velocity. We quantify this conflict using the *Local Guidance-to-Structure Ratio*:

$$\rho_{i,j} = \frac{w\|\hat{\delta v}_{i,j}\|_2}{\|v_{i,j}(x)\|_2 + \epsilon}. \quad (5)$$

Here, the denominator $\|v(x)\|$ acts as the "velocity foundation," while the numerator represents the "modification intensity." A dangerously high ratio implies the model is forcing strong changes in an area with low velocity support. To prevent local collapse, we adaptively suppress the guidance with hyperparameter $\gamma$:

$$\delta v_{i,j}^{ST} = \hat{\delta} v_{i,j} \cdot \text{clip}\left(\frac{\gamma}{\rho_{i,j}}, 0, 1\right). \tag{6}$$

As shown in Figure 4, this acts as a safety anchor: it dampens the guidance only in unstable regions where the external force exceeds the structural capacity, ensuring clean results without limiting creativity in feature-rich areas.

In summary, **Temporal Decay** is motivated by ODE stability: large step-to-step spikes in guidance can make the dynamics unstable or oscillatory, so damping them enforces practical temporal smoothness. **Spatial Suppression** is motivated by a local relative-perturbation principle: if the guidance becomes too large relative to the unconditional velocity at a location, that region is more likely to leave the stable transport neighborhood and create artifacts.

### 4.3. Summary: The MIST Transform

Integrating these components, the final **MIST** update rule is formulated as a nested refinement:

$$v_t^{MIST} = \text{IA}\left(v_t(x) + w \cdot \text{ST}(\delta v_t)\right). \tag{7}$$

It follows a Locally-Regulated, Globally-Aligned paradigm:

1. **Local Stage (ST):** First, ST acts as a **dynamical filter** on the raw guidance $\delta v_t$, suppressing local singularities and energy spikes arising from the quadratic instability.

2. **Global Stage (IA):** Subsequently, IA serves as a **statistical wrapper**, removing the linear barycentric drift and normalizing the global kinetic energy to match the prior distribution.

Unlike standard CFG which performs unconstrained extrapolation, MIST enforces rigorous geometric constraints, enabling high-fidelity generation under wide guidance scales ($w \in [2, 20]$). See appendix for the performance curve.

## 5. Experiment

### 5.1. Experimental Setup

**Implementation Details** We use the official model and implementations for each baseline and base model. All comparisons are conducted under a standardized setup for each base model: same base checkpoint, same prompts/seeds, same resolution, same number of sampling steps, same solver, and no extra model evaluations or denoising steps for any method. For all baseline hyperparameters, we follow

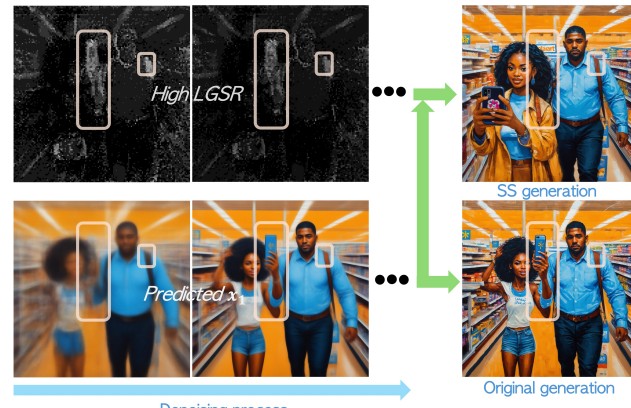

*Figure 4.* Spatial Suppression (SS). High Local Guidance-to-Structure Ratio (LGSR) often corresponds to artifact-prone regions. By adaptively clipping the guidance, SS suppresses these singularities while preserving high-frequency details in structurally rich areas, eliminating artifacts. Prompt: *A painting depicting a black woman taking a selfie in Wal-Mart while being followed by a man.*

official implementations or author-recommended settings. Specific details are provided in the appendix.

**Baselines and Base models.** We compare standard CFG and three advanced guidance methods: APG (Sadat et al., 2025), CFG++ (Chung et al., 2025), and CFG-Zero (Fan et al., 2025), where CFG-Zero is also designed for flow-based models. We map the guidance scale for CFG++ to its hyperparameter. For T2I base models, we employ large-scale flow-based models including Stable Diffusion 3 medium (SD3) (Esser et al., 2024), SD3.5 medium (Esser et al., 2024), Lumina-Next (Zhuo et al., 2024), and Flux-dev (Labs, 2024; Labs et al., 2025). The main experiments and ablations are based on the SD3.5 medium model. Please note that Flux-dev is a CFG-distilled model. We employ different guidance scales to mimic its CFG mechanism. For T2V tasks, we utilize the latest state-of-the-art Wan2.2 5B and Wan2.2 A14B models (Wan et al., 2025).

**Benchmarks.** Our evaluation contain both text-to-image (T2I) and text-to-video (T2V) tasks. For T2I evaluation, we use three prominent benchmarks: HPD v2 (Wu et al., 2023), which comprises 3,200 prompts across four styles (animation, concept art, paintings, and photos); GenEval (Ghosh et al., 2023), which focuses on object-centric text-to-image generation using compositional prompts to assess the model's understanding of complex relationships; and DPG (Hu et al., 2024), which consists of 1K dense prompts, enabling fine-grained assessment of different aspects of prompt adherence. These benchmarks are designed to assess model performance in complex scenes. For T2V evaluation, we adopt the standard prompts and evaluation metrics provided by VBench (Huang et al., 2024), which contains around 1K prompts for different dimensions.

**Metrics.** For the standard GenEval, DPG, and VBench

*Table 1.* Quantitative comparisons on HPD v2 benchmark. "G.S." is guidance scale; "P.S." is PickScore; "Aes." is aesthetic; "I.R." is ImageReward; "U.R." is UnifiedReward.

| Model | G.S. | P.S. | Aes. | CLIP | HPS | I.R. | U.R. |
|---|---|---|---|---|---|---|---|
| CFG (Ho & Salimans, 2022) | | 22.78 | 5.984 | 37.03 | 29.66 | 1.0818 | 3.3988 |
| CFG++ (Chung et al., 2025) | | 20.93 | 5.814 | 32.75 | 23.98 | -0.0159 | 2.5197 |
| APG (Sadat et al., 2025) | 5.0 | 21.82 | 5.985 | 35.06 | 24.81 | 0.4964 | 2.9532 |
| CFG-Zero (Fan et al., 2025) | | 22.84 | 6.014 | 36.89 | **30.31** | 1.0876 | 3.4190 |
| **MIST** | | **22.95** | **6.022** | **37.26** | 30.22 | **1.1126** | **3.4230** |
| CFG (Ho & Salimans, 2022) | | 22.44 | 5.866 | 36.57 | 29.21 | 1.0361 | 3.3662 |
| CFG++ (Chung et al., 2025) | | 22.26 | 6.020 | 36.37 | 28.23 | 0.8044 | 3.1727 |
| APG (Sadat et al., 2025) | 10.0 | 22.42 | 6.040 | 36.24 | 27.37 | 0.8606 | 3.2494 |
| CFG-Zero (Fan et al., 2025) | | 22.72 | 5.972 | 37.00 | 30.64 | 1.1558 | 3.4431 |
| **MIST** | | **23.02** | **6.053** | **37.33** | **31.29** | **1.2216** | **3.4958** |
| CFG (Ho & Salimans, 2022) | | 21.43 | 5.507 | 34.31 | 25.15 | 0.4922 | 2.9521 |
| CFG++ (Chung et al., 2025) | | 22.60 | 6.051 | 36.98 | 29.76 | 1.0092 | 3.3510 |
| APG (Sadat et al., 2025) | 15.0 | 22.67 | 6.065 | 36.67 | 28.68 | 0.9934 | 3.3685 |
| CFG-Zero (Fan et al., 2025) | | 22.25 | 5.824 | 36.60 | 29.27 | 1.0363 | 3.2907 |
| **MIST** | | **22.98** | **6.067** | **37.31** | **31.60** | **1.2482** | **3.4812** |

*Table 2.* Ablation study on IA (Invariant Alignment) and ST (Stability Thresholding). Both components bring improvements.

| Model | P.S. | Aes. | CLIP |
|---|---|---|---|
| CFG | 22.44 | 5.866 | 36.57 |
| +IA | 22.92 | 6.032 | 37.16 |
| +ST | 22.72 | 5.950 | 37.22 |
| +Both | **23.02** | **6.053** | **37.33** |

*Table 3.* Ablation study on ST strategies. "SS" is spatial suppression and "TM" is temporal decay.

| Model | P.S. | Aes. | CLIP |
|---|---|---|---|
| CFG | 22.44 | 5.866 | 36.57 |
| +SS | 22.71 | 5.935 | 37.21 |
| +TM | 22.61 | 5.929 | 36.94 |
| +Both | **22.72** | **5.950** | **37.22** |

*Table 4.* Quantitative comparisons on DPG benchmark (Hu et al., 2024). "Attr." refers to "Attribute". MIST achieves state-of-the-art results on the overall metric.

| Model | G.S. | Global | Entity | Attr. | Relation | Other | Overall |
|---|---|---|---|---|---|---|---|
| CFG (Ho & Salimans, 2022) | | 84.50 | 90.27 | **88.38** | 93.65 | 82.80 | 84.36 |
| CFG++ (Chung et al., 2025) | | 79.79 | 82.89 | 80.39 | 90.35 | 70.40 | 74.70 |
| APG (Sadat et al., 2025) | 5.0 | 82.98 | 86.89 | 85.11 | 92.22 | 75.60 | 80.03 |
| CFG-Zero (Fan et al., 2025) | | 84.50 | 90.48 | 88.28 | 93.60 | 81.90 | 84.97 |
| **MIST** | | **85.11** | **90.64** | 88.37 | **93.71** | **82.90** | **85.16** |
| CFG (Ho & Salimans, 2022) | | 82.29 | 90.49 | 88.23 | 93.49 | 83.70 | 84.51 |
| CFG++ (Chung et al., 2025) | | 84.80 | 87.57 | 85.76 | 91.78 | 78.00 | 81.67 |
| APG (Sadat et al., 2025) | 10.0 | **85.33** | 89.22 | 86.97 | 93.38 | 79.10 | 83.09 |
| CFG-Zero (Fan et al., 2025) | | 83.43 | 90.92 | 88.62 | 93.82 | 82.30 | 85.29 |
| **MIST** | | 84.19 | **91.41** | **88.64** | **94.13** | **85.20** | **85.87** |
| CFG (Ho & Salimans, 2022) | | 78.34 | 87.44 | 84.52 | 91.59 | 80.10 | 79.93 |
| CFG++ (Chung et al., 2025) | | **85.94** | 88.37 | 86.60 | 92.28 | 79.50 | 82.96 |
| APG (Sadat et al., 2025) | 15.0 | 85.26 | 90.01 | 87.70 | 93.55 | 80.50 | 84.31 |
| CFG-Zero (Fan et al., 2025) | | 81.00 | 89.85 | 87.51 | 93.16 | 82.00 | 83.40 |
| **MIST** | | 83.97 | **91.83** | **88.58** | **94.55** | **85.70** | **86.40** |

*Table 5.* Comparisons on SD3 base model.

| Model | G.S. | P.S. | Aes. | CLIP | HPS |
|---|---|---|---|---|---|
| CFG | 5.0 | 22.64 | 5.956 | **36.41** | 29.64 |
| **MIST** | | **22.73** | **5.998** | 36.34 | **29.84** |
| CFG | 10.0 | 22.35 | 5.845 | 36.53 | 29.53 |
| **MIST** | | **22.69** | **5.988** | **36.86** | **30.68** |
| CFG | 15.0 | 21.73 | 5.606 | 35.80 | 27.54 |
| **MIST** | | **22.59** | **5.971** | **36.88** | **30.65** |

*Table 6.* Comparisons on Lumina-Next.

| Model | G.S. | P.S. | Aes. | CLIP | HPS |
|---|---|---|---|---|---|
| CFG | 5.0 | 22.28 | 6.175 | 34.18 | 27.44 |
| **MIST** | | **22.50** | **6.255** | **34.40** | **28.13** |
| CFG | 10.0 | 21.65 | 5.972 | 33.41 | 26.08 |
| **MIST** | | **22.52** | **6.231** | **34.88** | **28.74** |
| CFG | 15.0 | 21.15 | 5.822 | 32.60 | 24.97 |
| **MIST** | | **22.47** | **6.212** | **35.01** | **28.86** |

benchmarks, we employ their official metrics. For HPD v2, we employ four types of overall human preference metrics: PickScore (Kirstain et al., 2023), HPSv2.1 (Wu et al., 2023), ImageReward (Xu et al., 2023), and UnifiedReward (Wang et al., 2025), where UnifiedReward is based on a state-of-the-art VLM model (Bai et al., 2025). Furthermore, we use the Aesthetic score (Schuhmann, 2022) and CLIP score (Radford et al., 2021) to measure the aesthetic quality and prompt-following ability, respectively.

## 5.2. Quantitative Evaluation

Table 1 presents the quantitative results of our proposed MIST compared to CFG across various methods on HPD v2 benchmark under different guidance scales (G.S.). MIST consistently achieves superior performance. As evidenced by the table, MIST surpasses other methods in terms of Aesthetic Score (Aes.), PickScore (P.S.), HPS, Image Re-

ward (I.R.), and Unified Reward (U.R.) across all guidance scales. Specifically, for a guidance scale (G.S.) of 15.0, MIST significantly improves the CLIP score to 37.31 and the Aesthetic score to 6.067, demonstrating its effectiveness in enhancing both text-image alignment and visual appeal. It also outperforms other methods on human preference metrics: PickScore, HPS, Image Reward (I.R.), and Unified Reward (U.R.). Moreover, the enhanced CLIP Score confirms that generated images better capture the semantics of the given prompts. The performance on other official benchmarks: GenEval (Table 7) and DPG (Table 4) further highlights the strength of our approach, showing its effectiveness in handling complex generation tasks and refining suboptimal results produced by CFG. Due to the space limitation, please refer to the appendix or results on Flux-dev.

We also conduct experiments on the text-to-video (T2V) generation task, which is a more challenging setting that re-

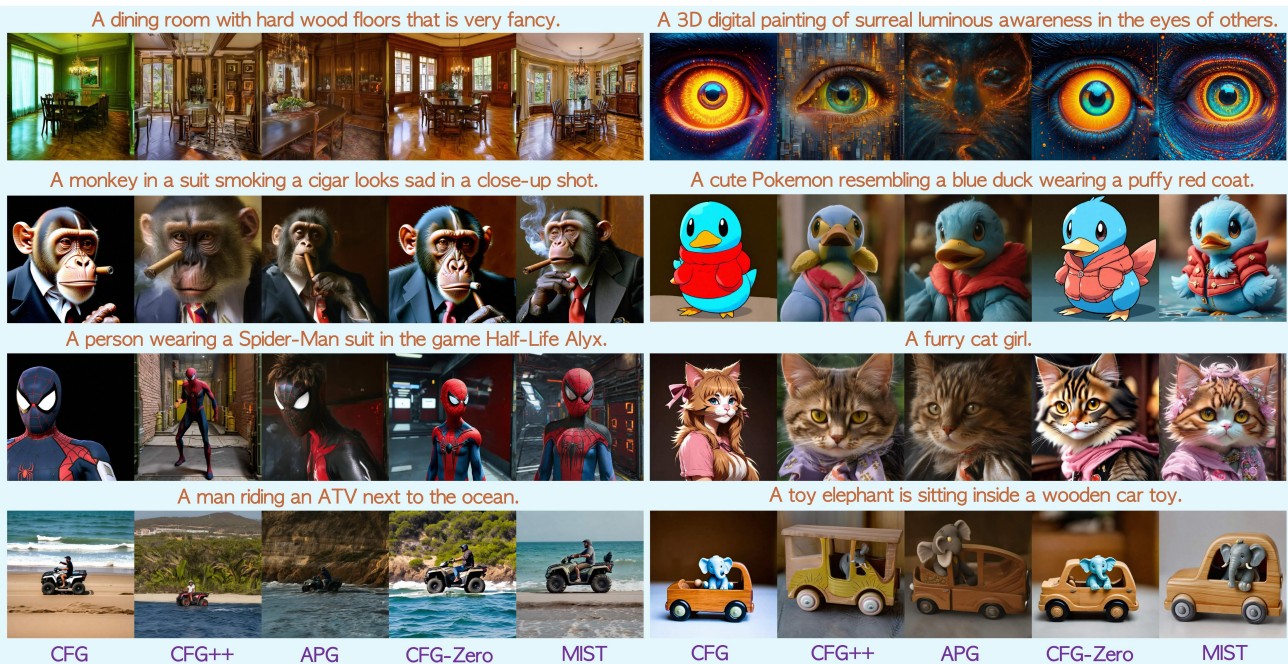

*Figure 5.* Qualitative comparisons on SD3.5 medium base model at guidance scale 10. MIST obtains more visually appealing and superior prompt alignment.

*Table 7.* Quantitative comparisons on GenEval benchmark (Ghosh et al., 2023).

| Methods | G. S. | Single Object | Two Object | Counting | Colors | Position | Color Attribution | Overall |
|---|---|---|---|---|---|---|---|---|
| CFG (Ho & Salimans, 2022) | | 99.38 | 83.08 | **65.00** | 81.12 | 23.75 | 47.15 | 66.58 |
| CFG++ (Chung et al., 2025) | | 76.25 | 39.65 | 25.31 | 45.48 | 8.75 | 13.21 | 34.78 |
| APG (Sadat et al., 2025) | 5.0 | 95.94 | 64.65 | 39.38 | 71.54 | 14.50 | 27.85 | 52.31 |
| CFG-Zero (Fan et al., 2025) | | 99.69 | 82.58 | 60.94 | **83.51** | 24.00 | **49.39** | 66.68 |
| **MIST** | | **100.00** | **85.10** | 63.44 | 82.18 | **25.50** | 46.14 | **67.09** |
| CFG (Ho & Salimans, 2022) | | 99.06 | **88.64** | **66.88** | 78.19 | 27.25 | 43.09 | 67.18 |
| CFG++ (Chung et al., 2025) | | 94.38 | 73.23 | 44.06 | 72.07 | 21.25 | 37.40 | 57.07 |
| APG (Sadat et al., 2025) | 10.0 | 99.38 | 73.99 | 50.94 | 78.72 | 19.25 | 38.21 | 60.08 |
| CFG-Zero (Fan et al., 2025) | | 99.38 | 85.61 | 64.38 | **82.71** | 25.25 | 46.54 | 67.31 |
| **MIST** | | **100.00** | 87.12 | 62.81 | 80.32 | **27.64** | **52.44** | **68.39** |
| CFG (Ho & Salimans, 2022) | | 95.31 | 81.06 | 54.69 | 71.01 | 22.25 | 30.69 | 59.17 |
| CFG++ (Chung et al., 2025) | | 97.81 | 80.56 | 54.37 | 80.59 | 25.00 | 41.46 | 63.30 |
| APG (Sadat et al., 2025) | 15.0 | 99.06 | 81.06 | 50.94 | 81.12 | 22.00 | 44.72 | 63.15 |
| CFG-Zero (Fan et al., 2025) | | 99.69 | 84.60 | 61.88 | 78.19 | 25.25 | 40.04 | 64.94 |
| **MIST** | | **100.00** | **86.36** | **64.06** | **83.51** | 26.44 | 50.20 | **68.43** |

quires both spatial fidelity and temporal coherence. The quantitative results for text-to-video generation are presented in Table 8. Specifically, when applied to the Wan2.2 model (which includes the 5B and A14B versions) (Wan et al., 2025), MIST demonstrates marked improvements across several key metrics.

### 5.3. Qualitative Evaluation

The qualitative comparisons are presented in Figure 5 (and Figure 6 for video), offering a comprehensive visual demonstration of our method's efficacy. MIST consistently produces high-quality images, characterized by rich detail and strong semantic alignment with the given text descriptions. Furthermore, when applied to video generation, MIST yields more temporally consistent and coherent frames, mitigating artifacts often observed in other methods.

### 5.4. Ablations

We provide detailed ablations on different components of MIST. We employ SD3.5 medium base model and a guidance scale of 10 by default for these ablation studies. More ablations are provided in the supplementary material.

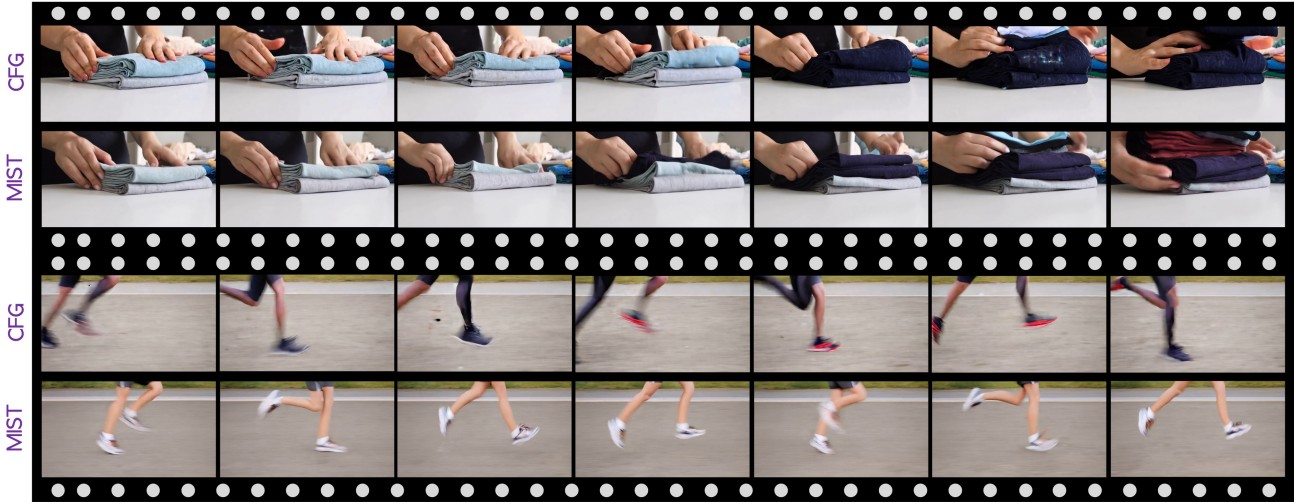

*Figure 6.* Qualitative comparisons on Wan2.2 5B base model at guidance scale 9. MIST obtains more consistent and coherent results.

*Table 8.* Comparisons on Vbench benchmark. We use the recent Wan2.2 models as our base model. Compared to vanilla CFG, MIST improves both frame aesthetics and overall video quality.

| Model | Guidance | Aesthetic Quality | Motion Smoothness | Overall Consistency | Spatial Relationship | Temporal Style | Quality Score | Semantic Score | Total Score |
|---|---|---|---|---|---|---|---|---|---|
| | CFG 4.0 | 58.69 | **98.69** | 24.81 | 75.38 | 24.81 | 83.02 | 71.19 | 80.65 |
| Wan2.2 5B (Wan et al., 2025) | CFG 9.0 | 59.09 | 98.22 | 25.36 | **80.67** | 24.82 | 83.36 | **74.74** | 81.64 |
| | **MIST 9.0** | **59.69** | 98.53 | **25.55** | 80.15 | **25.02** | **83.89** | 74.05 | **81.92** |
| | CFG 4.0 | 62.69 | 98.20 | 26.14 | 79.86 | 23.92 | 83.93 | 75.81 | 82.30 |
| Wan2.2 A14B (Wan et al., 2025) | CFG 9.0 | 62.64 | 97.73 | 26.23 | **80.95** | **24.26** | 83.63 | 76.66 | 82.24 |
| | **MIST 9.0** | **62.82** | **98.23** | **26.24** | 80.54 | 24.13 | **84.07** | **76.76** | **82.61** |

**Impact of IA (Invariant Alignment) and ST (Stability Thresholding).** Table 2 provides the ablation results for our two main components: IA and ST. Each component brings significant improvements. For example, IA improves Aesthetic score from 5.866 to 6.032, and ST boosts CLIP score from 36.57 to 37.22. When using both components, our method obtains the best performance across all metrics.

**Impact of IA strategies.** Table 9 provides the ablation of IA components. We observe that DC alone contributes margin improvements. The full IA (DC+EM) approach further enhances performance.

*Table 9.* Ablation study on IA. "DC" refers to drift correction and "EM" is energy matching.

| Model | P.S. | Aes. | CLIP |
|---|---|---|---|
| CFG | 22.44 | 5.866 | 36.57 |
| +DC | 22.58 | 5.903 | 36.91 |
| +DC, EM | **22.92** | **6.032** | **37.16** |

**Impact of ST strategies.** In Table 3, we show the ablation of different components in ST. SS makes the velocity more stable and suppresses outliers in a fine-grained manner. It brings significant improvements. TD facilitates the overall denoising dynamics, which further boosts the results.

## 6. Limitation and Future Work

While MIST is simple and consistently improves upon standard CFG across various models and benchmarks, it has several limitations. First, our analysis and design are based on the velocity-prediction framework used in flow-matching models. Although we demonstrate gains on popular flow-based models (*e.g.*, SD3, SD3.5, Lumina-Next, Flux, Wan2.2), MIST has not been exhaustively evaluated on SDE-based models (*e.g.*, DDPM/EDM). Extending MIST to such frameworks may require additional adaptations. Second, our argument for stabilizing velocity distributions is empirical and intuitive; formal theoretical analysis of guidance-induced distribution shifts in deterministic samplers remains an open direction for future work.

## 7. Conclusion

In this work, we introduce MIST (Moment-aligned Invariant Stability Transform), a training-free method designed to resolve the inherent instabilities of Classifier-Free Guidance (CFG) in flow-matching models. By formalizing the CFG-induced shift via statistical moment decomposition, we identify that generation failures stem from two distinct sources: linear barycentric drift and quadratic energetic expansion.

To address these, MIST introduces a hierarchical intervention strategy: (1) Invariant Alignment (IA), comprising Drift Correction (DC) and Energy Matching (EM), which restores the global statistical integrity of the transport velocity; and (2) Stability Thresholding (ST), incorporating Temporal Decay (TD) and Spatial Suppression (SS), which enforces local Lipschitz continuity and suppresses numerical singularities. Extensive experiments across Text-to-Image (T2I) and Text-to-Video (T2V) tasks demonstrate that MIST consistently outperforms standard CFG and existing correction methods. Evaluations on benchmarks including HPD v2, GenEval, DPG, and VBench confirm significant gains in aesthetic quality, prompt alignment, and structural robustness. As a plug-and-play solution, MIST offers a versatile pathway to unlock the full potential of high-guidance scales in future flow-based generative models.

## Impact Statement

This paper presents MIST, a method to enhance the stability and fidelity of flow-based generative models. Positively, MIST empowers users with precise creative control, reducing trial-and-error costs and democratizing high-quality content creation. Regarding risks, enhanced realism could be misused for misinformation or harmful content. However, MIST is a sampling-time intervention fully compatible with existing safety protocols, such as dataset filtering, safety checkers, and watermarking. We advocate for responsible deployment with rigorous guardrails to mitigate potential misuse.

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

# Appendix

Due to the space limitation, we provide details omitted in the main text in this appendix and the supplementary material, which is organized as follows:

- Section A : Algorithm overview.

- Section B : Comparisons on different T2I base models.

- Section C : Ablations on hyperparameters.

- Section D : Detailed implementation for different models and baselines.

- Section E : Robustness over wide guidance range.

- Section F : More visualization on T2I base models.

- Section G : Failure cases.

For better visualization of video results, please refer to `Visualization_webpage.html` in the supplementary material.

## A. Algorithm Overview

We provide an overview of MIST in Algorithm 1. It is a plug-and-play module and can be easily implemented for current flow-based diffusion models.

---

**Algorithm 1** The Proposed Guidance Method: MIST.

---

1: **Input:** Velocity prediction $v_t(x), v_t(x|y)$, guidance scale $w$, clipping factor $\gamma$, start step $T_{\text{clip}}$, total steps $T$.
2: Initialize $\|\delta v_{t+1}\| \leftarrow 0$, step counter $cnt \leftarrow 0$
3: **for** $t = T$ to $0$ **do**
4:    Compute raw guidance: $\delta v_t = v_t(x|y) - v_t(x), \quad cnt \leftarrow cnt + 1$
5:    # 1. Stability Thresholding (ST)
6:    **if** $cnt > T_{\text{clip}}$ **then**
7:       // Temporal Decay (TD)
8:       $r_t \leftarrow \min\left(1, \frac{\|\delta v_{t+1}\|_2}{\|\delta v_t\|_2 + \epsilon}\right)$
9:       $\delta v_t \leftarrow \delta v_t \cdot r_t$
10:       // Spatial Suppression (SS)
11:       $\rho_{i,j} \leftarrow \frac{w\|\delta v_t^{i,j}\|_2}{\|v_t^{i,j}(x)\|_2 + \epsilon}$
12:       $\delta v_t^{i,j} \leftarrow \delta v_t^{i,j} \cdot \text{clip}\left(\frac{\gamma}{\rho_{i,j}}, 0, 1\right)$
13:    **end if**
14:    Update history: $\|\delta v_{t+1}\| \leftarrow \|\delta v_t\|_2$
15:    # 2. Invariant Alignment (IA)
16:    // Drift Correction (DC)
17:    $\delta v_t^{dc} \leftarrow \delta v_t - \mathbb{E}[\delta v_t]$
18:    $v_t^{dc} \leftarrow v_t(x) + w \cdot \delta v_t^{dc}$
19:    // Energy Matching (EM)
20:    $\mu \leftarrow \mathbb{E}[v_t(x)], \quad \sigma_{src} \leftarrow \text{std}(v_t^{dc}), \quad \sigma_{tgt} \leftarrow \text{std}(v_t(x))$
21:    $v_t^{\text{MIST}} \leftarrow \mu + (v_t^{dc} - \mu) \cdot \frac{\sigma_{tgt}}{\sigma_{src} + \epsilon}$
22:    # 3. ODE Solver Step
23:    $x_{t-1} \leftarrow \text{ODEStep}(v_t^{\text{MIST}}, x_t)$
24: **end for**
25: **Return** clean latent $x_0$

---

## B. Comparisons on Other T2I Base Models

The main experiments and ablations are based on SD3.5 medium model (Esser et al., 2024). We also verify our method on other flow-based models, including Stable Diffusion 3 medium (SD3) (Esser et al., 2024) and Lumina-Next (Zhuo et al., 2024) in the main text. Here we provide comparisons on Flux-dev (Labs, 2024; Labs et al., 2025). As shown in Table 10, MIST shows consistent improvements.

*Table 10.* Comparisons on Flux-dev base model. Please note that Flux-dev is a CFG-distilled model. For a fair comparison, we mimic the guidance mechanism.

| Model | Guidance Scale | PickScore | Aesthetic | CLIP | HPS | ImageReward | UnifiedReward |
|-------|---------------|-----------|-----------|-------|-------|-------------|---------------|
| CFG | 5.0 | 22.87 | 6.009 | 36.88 | 28.90 | 1.1284 | 3.4244 |
| **MIST** | | **23.07** | **6.092** | **37.01** | **29.76** | **1.1827** | **3.4476** |
| CFG | 10.0 | 22.22 | 5.661 | 35.63 | 26.87 | 0.8747 | 3.1173 |
| **MIST** | | **23.03** | **6.071** | **37.29** | **30.25** | **1.2238** | **3.4506** |
| CFG | 15.0 | 21.36 | 5.244 | 33.06 | 23.32 | 0.3743 | 2.6609 |
| **MIST** | | **22.98** | **6.062** | **37.35** | **30.42** | **1.2363** | **3.4340** |

## C. Ablations on Hyperparameter

In this section, we investigate the impact of hyperparameters in MIST. For $T_{clip}$, it controls the timesteps of applying the proposed Temporal Decay (TD) strategy. As shown in Table 11, we can obtain the best aesthetic score when employing TD all the time, while it sacrifices the CLIP score because the first several denoising steps are unstable. Thus we set $T_{clip}$ to 1 by default for the model of 28 sampling steps. For $\gamma$ in Spatial Suppression (SS), it determines the clipping norm threshold in the spatial clipping part. As shown in Table 12, higher value refers to lower norm, and it will more aggressively clip the norm. We find it is beneficial for semantics, and we set $\gamma = 1.5$ to achieve performance balance.

## D. Implementation Details

### D.1. Base Models

For all base models, we generate $1024 \times 1024$ images, and we follow their official sampling settings, and here we describe related details. For **SD3/SD3.5**, we employ the medium models and use the same sampling setting, namely, denoising steps. For **Lumina-Next**, we employ the official model and use denoising steps 30. For **Flux-dev**, We employ the officially released model. Note that it is a CFG-distilled model, thus we modify its pipeline to mimic standard CFG. Specifically, we set `guidance_scale` as 1.0 and use the `true_cfg_scale` in the pipeline, to control its CFG scale. For **Wan2.2 5B** base model, note that it uses a new highly-compressed VAE, we use the recommended resolution of $121 \times 704 \times 1280$ (f, h, w). For **Wan2.2 A14B** base model, it uses a standard video VAE as Wan 2.1. Concerning its high computation cost, we use the recommended resolution of $81 \times 480 \times 832$ (f, h, w). We use the norm dimension 1 for temporal clipping and norm dimensions 3 and 4 for spatial clipping. Namely, we suppress outliers in a more fine-grained manner across subsequent denoising steps. We find that this manner is more stable for text-to-video models.

*Table 11.* Ablation study on $T_{\text{clip}}$.

| $T_{clip}$ | PickScore | Aesthetic | CLIP | HPS | ImageReward | UnifiedReward |
|-----------|-----------|-----------|-------|-------|-------------|---------------|
| 0 | 23.01 | **6.056** | 37.32 | 31.30 | 1.2200 | 3.4915 |
| 1 | **23.02** | 6.053 | **37.33** | 31.29 | 1.2216 | **3.4958** |
| 2 | 22.99 | 6.043 | 37.27 | 31.30 | 1.2228 | 3.4801 |
| 3 | 22.99 | 6.037 | 37.24 | 31.35 | 1.2264 | 3.4670 |
| 4 | 22.98 | 6.039 | 37.21 | 31.38 | 1.2251 | 3.4726 |
| 5 | 22.98 | 6.038 | 37.20 | **31.40** | **1.2310** | 3.4679 |

*Table 12.* Ablation study on $\gamma$.

| $\gamma$ | PickScore | Aesthetic | CLIP | HPS | ImageReward | UnifiedReward |
|---|---|---|---|---|---|---|
| 0.5 | 22.94 | 6.038 | 37.21 | 31.42 | 1.2299 | 3.4702 |
| 1.0 | 22.99 | 6.043 | 37.27 | **31.52** | **1.2321** | 3.4824 |
| 1.5 | **23.02** | **6.053** | 37.33 | 31.29 | 1.2216 | **3.4958** |
| 2.0 | **23.02** | 6.048 | 37.40 | 31.06 | 1.1982 | 3.4903 |
| 2.5 | 23.01 | 6.040 | **37.46** | 30.78 | 1.1718 | 3.4772 |
| 3.0 | 22.99 | 6.040 | **37.46** | 30.43 | 1.1450 | 3.4585 |

*Table 13.* Robustness on different sampling steps.

| Sampling steps | PickScore | Aesthetic | ImageReward |
|---|---|---|---|
| 28 | 22.44 | 5.866 | 1.0361 |
| + MIST | **23.02** | **6.053** | **1.2216** |
| 50 | 22.73 | 5.938 | 1.1720 |
| + MIST | **23.02** | **6.060** | **1.2369** |

*Table 14.* Robustness on different time spacing (time shift).

| Timeshift | PickScore | Aesthetic | ImageReward |
|---|---|---|---|
| 1 | 21.47 | 5.555 | 0.4172 |
| + MIST | **22.85** | **5.993** | **1.1349** |
| 3 | 22.44 | 5.866 | 1.0361 |
| + MIST | **23.02** | **6.053** | **1.2216** |

### D.2. Baselines

For different baselines, we follow their official implementations. **CFG++** does not require the guidance scale $w$ in the CFG. Instead, it employs a hyperparameter $(0.0 - 1.0)$ to implement guidance. To align other methods using standard CFG guidance, we map the $0 - 20$ guidance scale to $0.0 - 1.0$, which is the parameter required for CFG++. Moreover, to fit flow-based methods, we follow the authors' instructions in their official implementations [1]. For **APG**, we use the detailed implementation in their paper, with hyperparameters employed for DiT-XL/2, namely, $\eta = 0, r = 5, \beta = -0.5$. Please note that both CFG++ and APG are not designed for flow-based methods. For **CFG-Zero**, we directly adopt its official implementation [2] and use the default settings in SD3 pipeline.

## E. Robustness over Wide Guidance Range

Considering that MIST works well on high guidance scales, we investigate its robustness over a wide guidance range. As shown in Figure 7, the performance of CFG rapidly decreases at high guidance scales, while MIST works well across different guidance scales, demonstrating its robustness. We also perform experiments on different sampling steps and time spacing in Table 13 and Table 14.

## F. Visualization on More T2I Base Models

In the main text, we provide visualization results for SD3.5 medium base model. Here we provide results of other base models. For **Flux-dev** base model, we provide qualitative results in Figure 10. For **SD3 medium** base model, we provide qualitative results in Figure 11. For **Lumina-Next** base model, we provide qualitative results in Figure 12.

For the results in Figure 1 in the main text of **SD3.5 medium** base model, the prompts from top to bottom are:

*There is a white toilet and a sink in this bathroom.*

*A brown cat crouches and arches its back in a white sink.*

*A vase with a flower growing very well.*

*Professional digital art of Godzilla with stunning detail.*

*Two colorful parrots perched together eating an egg tart.*

*A miniature anthropomorphic cat knight wearing pale blue armor and a crown.*

---

[1]https://github.com/CFGpp-diffusion/CFGpp/issues/12

[2]https://github.com/WeichenFan/CFG-Zero-star

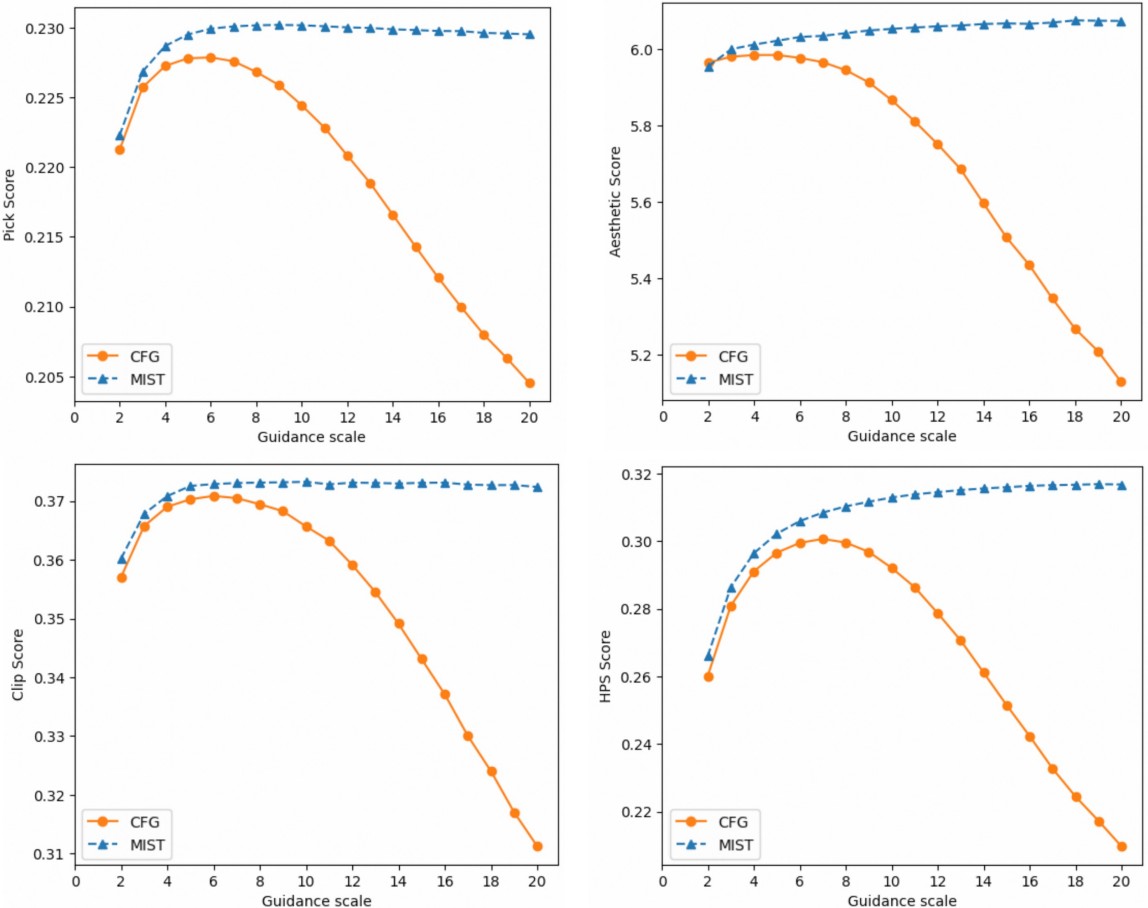

*Figure 7.* Results on the guidance scale from 2 to 20.

*A flat ink sketch of a hedgehog in the comic book style of Jim Lee.*

*Steve Buscemi portrays the Joker.*

## G. Failure cases

### G.1. Failure cases of MIST

MIST is most effective when the guidance scale is sufficiently high to induce instability. Its benefits may be less pronounced in the following situations: (1) under very low guidance scales, where there is little instability to correct; (2) in cases where stronger local suppression may slightly reduce stylized exaggeration in exchange for improved stability; and (3) for models whose guidance behavior has already been heavily regularized or distilled, leaving limited room for further improvement. These cases indicate diminishing returns rather than systematic degradation. We provide some cases in Figure 8.

### G.2. Failure modes of other base models

We observe similar failure modes across different flow-base models, which is plausible given that they share the same flow-matching sampling paradigm. This is also consistent with our cross-model results. We provide qualitative examples in Figure 9.

## Low CFG

A kangaroo wearing an orange hoodie and blue sunglasses stands on the grass in front of the Sydney Opera House, holding a sign that says Welcome Friends.

A close-up portrait of Sailor Moon standing in front of Russian panel houses.

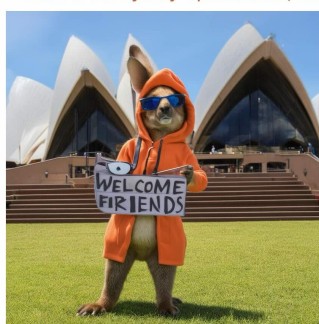 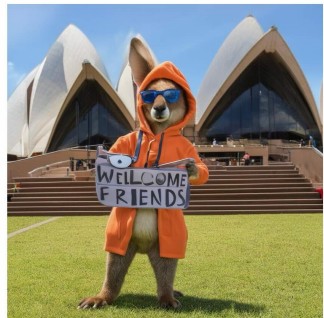 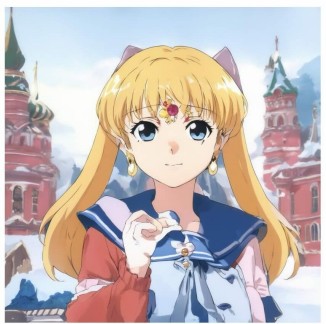 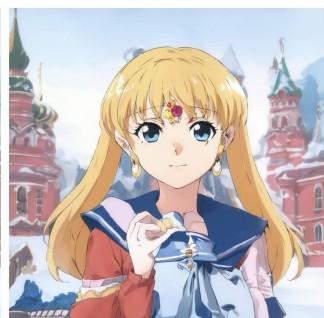

## Stylized exaggeration

Super Mario 64 level in Unreal Engine.

Realistic portrait painting of an astronaut suit with a 3D fractal lace design and iridescent bubble texture.

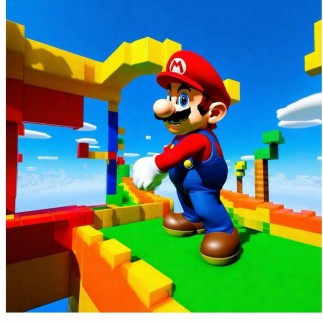 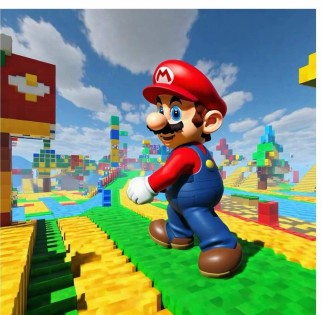 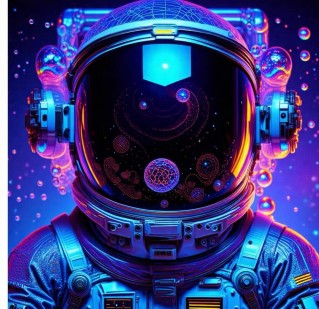 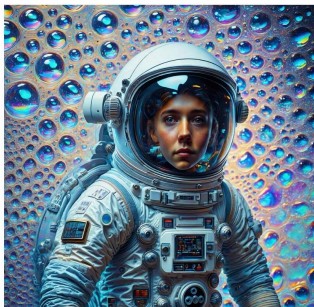

## w/ other CFG guidance methods

A photorealistic Bob Odenkirk is sitting under a tree with a smiling anime girl with black hair and hime cut in a digital art anime key visual.

There is a secret museum of magical items inside a crystal greenhouse pala ce filled with intricate bookshelves, plants, and Victorian style decor.

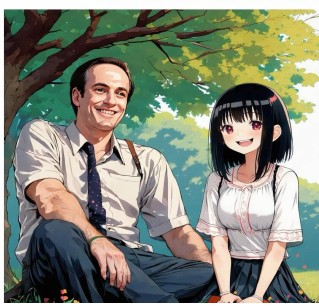 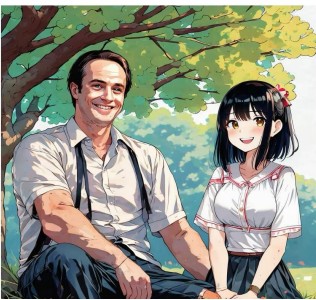 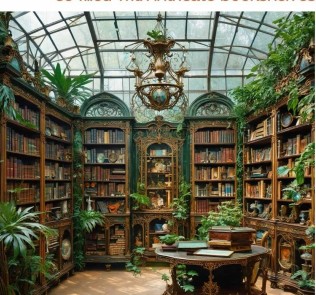 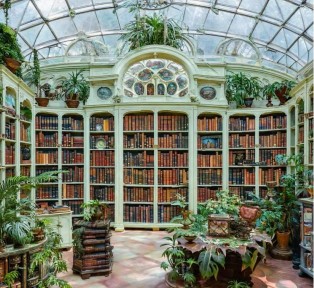

*Figure 8.* Failure cases of MIST. MIST may be less effective, or slightly harmful, in the following situations: First row, low CFG guidance scale (scale 2); second row, stylized exaggeration; third row, extension to other CFG-guidance methods.

Two motorcycles are parked on the shoulder of a mountainous freeway.

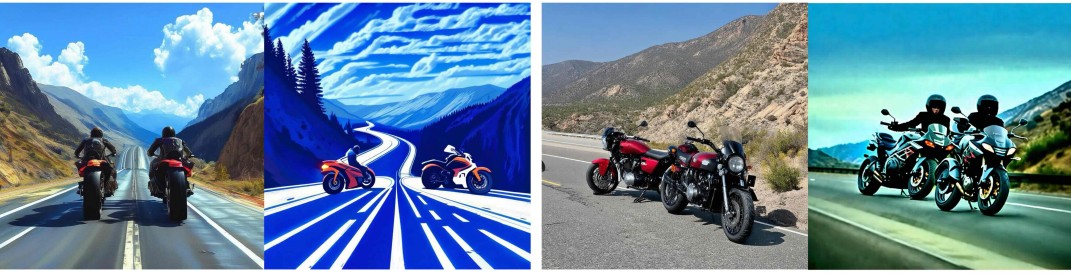

A dog looks out through a lined window.

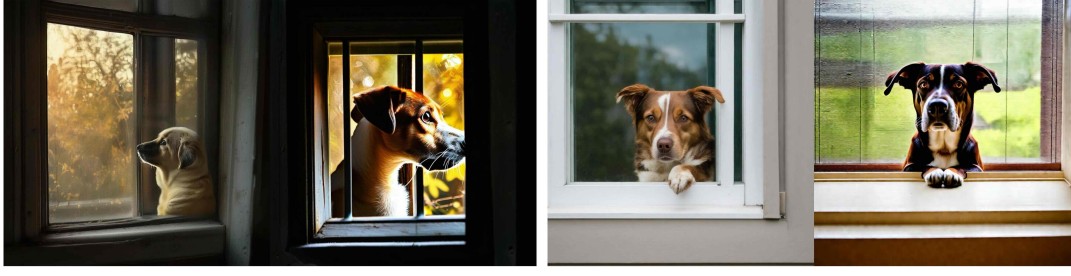

A bathroom with a toilet, sink and shower stall.

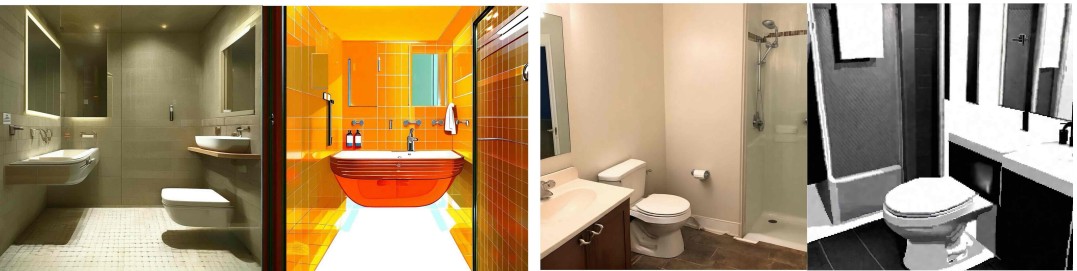

A group of people posing with festive items.

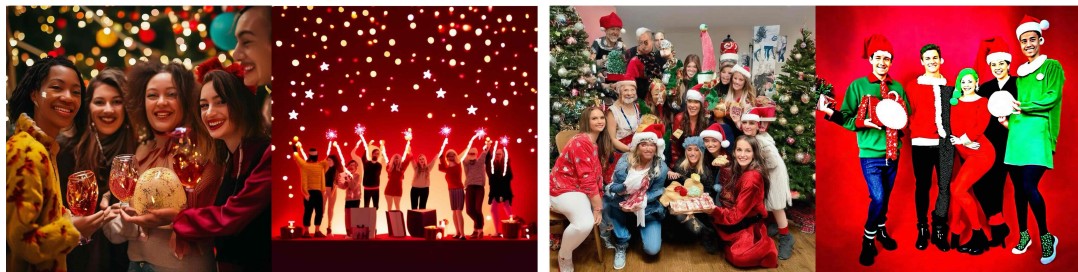

Fruit in a jar filled with liquid sitting on a wooden table.

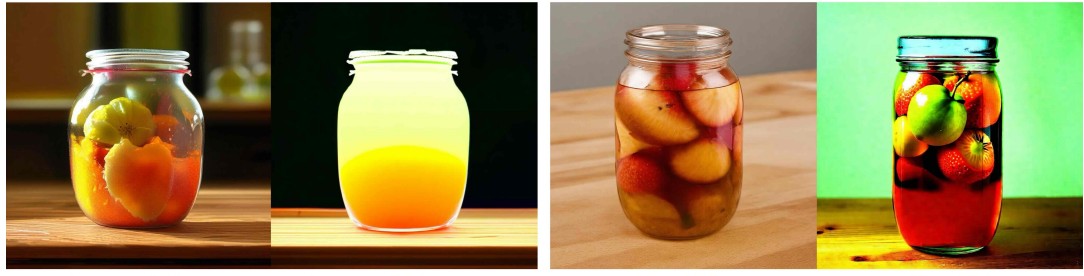

| Lumina CFG5 | Lumina CFG15 | SD3 CFG5 | SD3 CFG15 |

*Figure 9.* Failure cases on high CFG of other base models.

The image depicts a muscular mouse wielding assault rifles,
in a Disney art style.

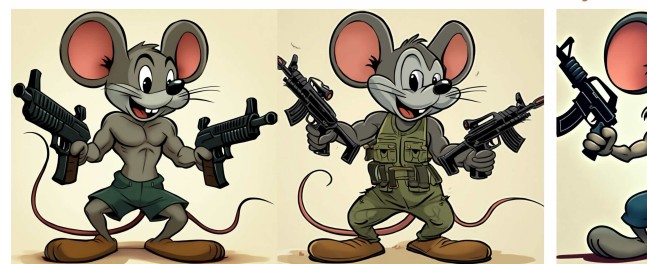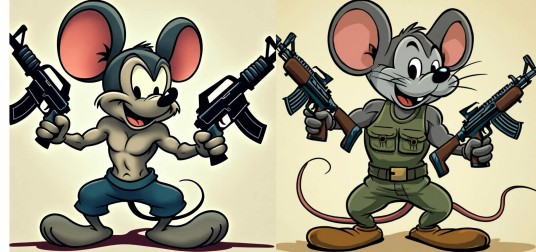

A dog resembling Hugh Laurie.

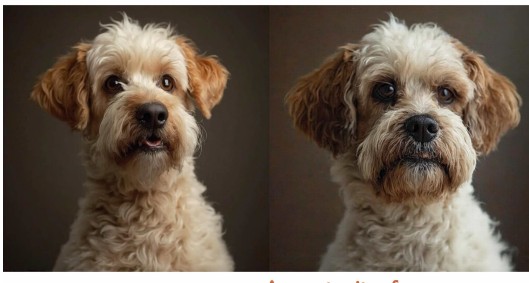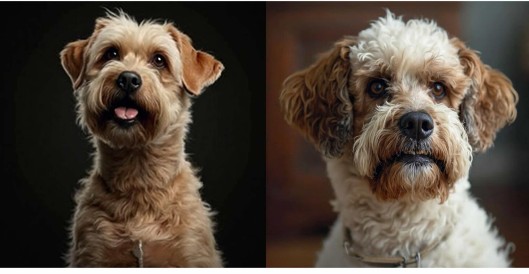

A portrait of a man resembling Super Mario against
a stylized background.

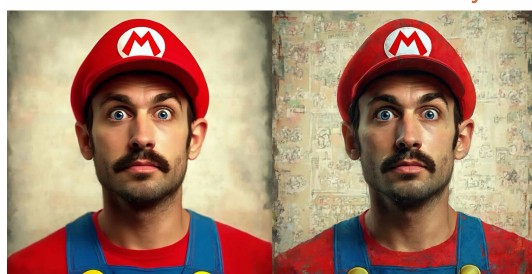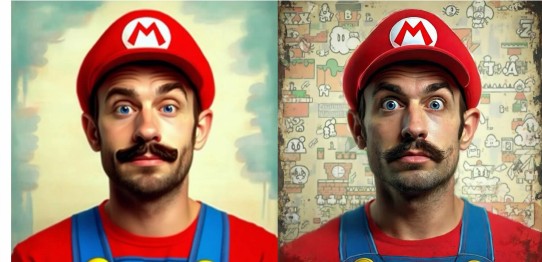

A digital painting of a favela city shrouded in mystical colors with radia
nt god rays and vibrant hues in the style of multiple artists.

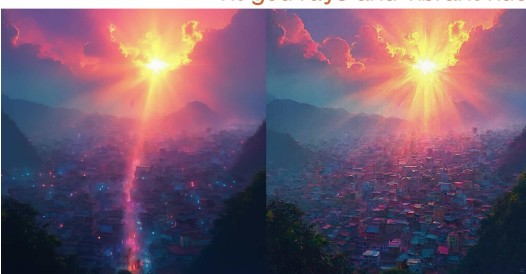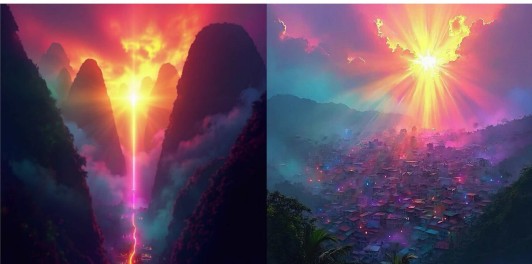

Animals fashioned from gems, colorful and shapely, depicted in natural
lighting, with a slight effervescence, artist credited to Alex Ross.

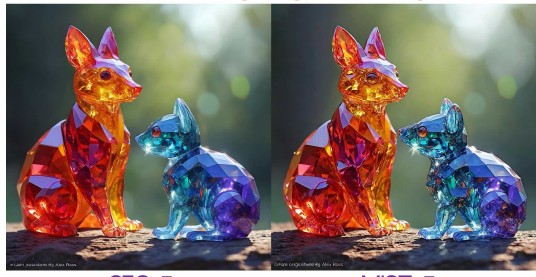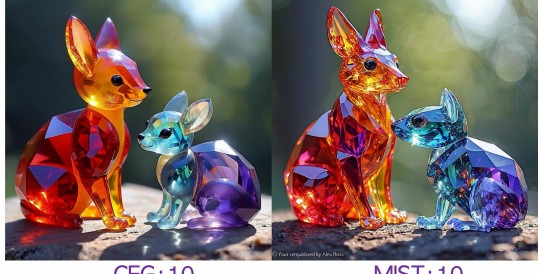

| CFG:5 | MIST:5 | CFG:10 | MIST:10 |

*Figure 10.* Qualitative results on Flux-dev base model.

Anime art featuring Hatsune Miku with symmetrical shoulders.

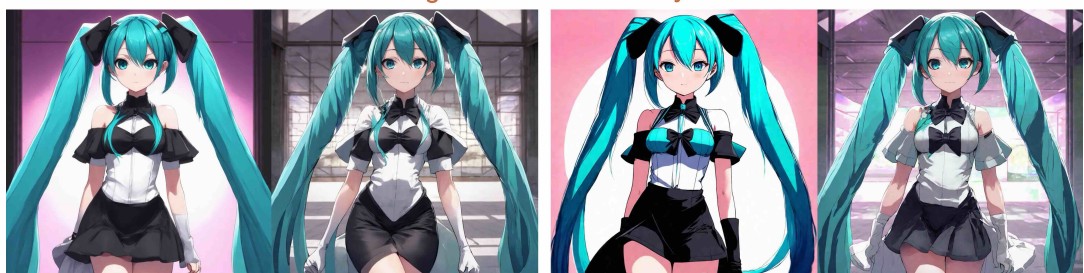

An illustration from the realistic comic book "Tiger White" featuring detailed artwork by a skilled illustrator.

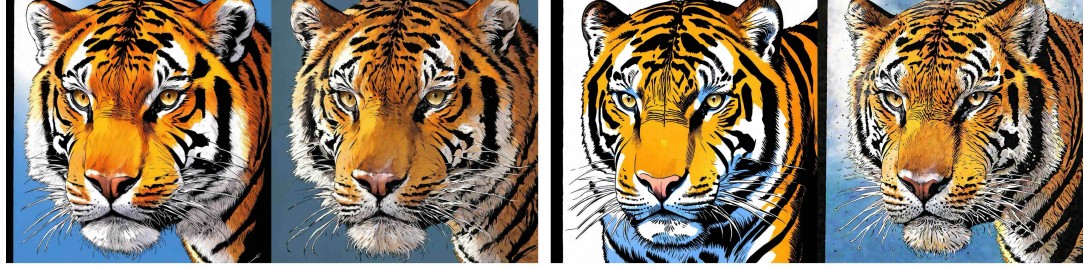

Doraemon is depicted as the Terminator using the Unreal Engine.

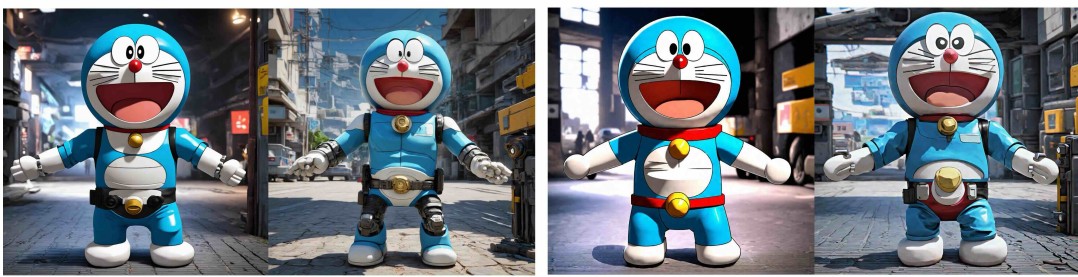

Darth Vader playing electric guitar on top of mountain.

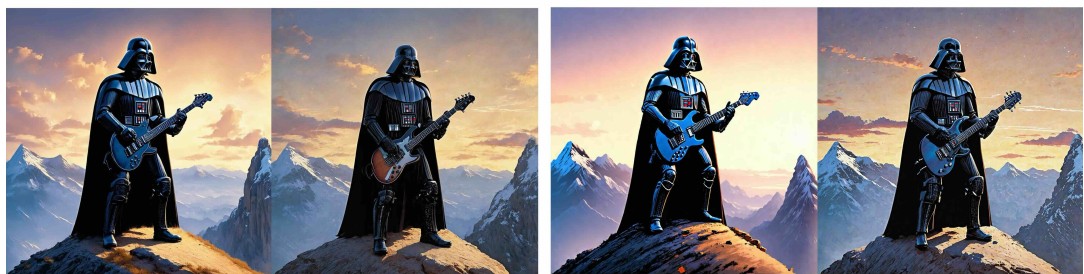

a man sitting on a motorcycle in the desert.

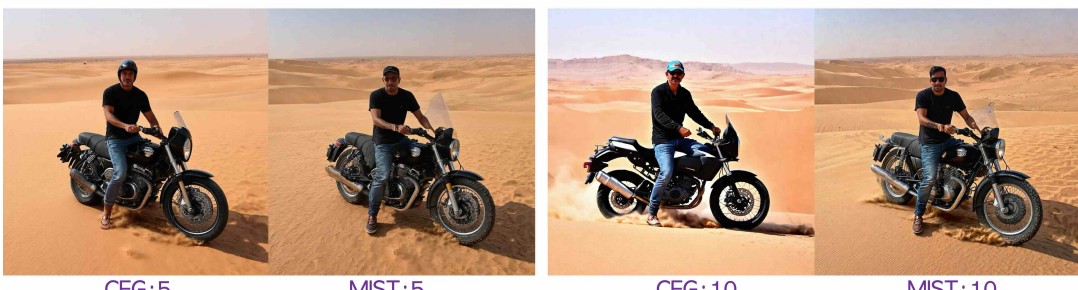

CFG:5      MIST:5      CFG:10      MIST:10

*Figure 11.* Qualitative results on SD3 medium base model.

A photo of Big Chungus from Looney Tunes.

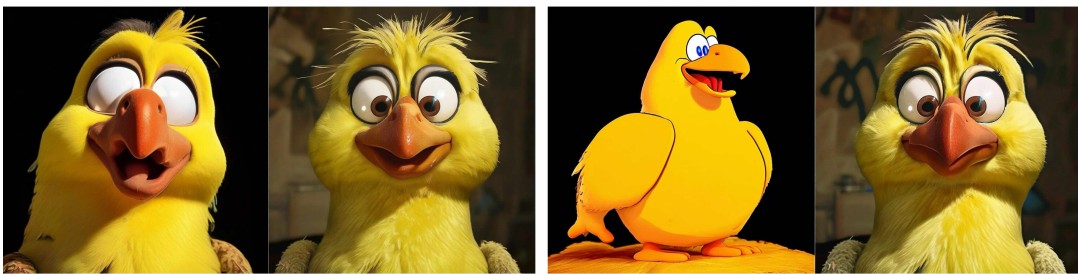

A man wearing a Batman costume holds a green glowing orb.

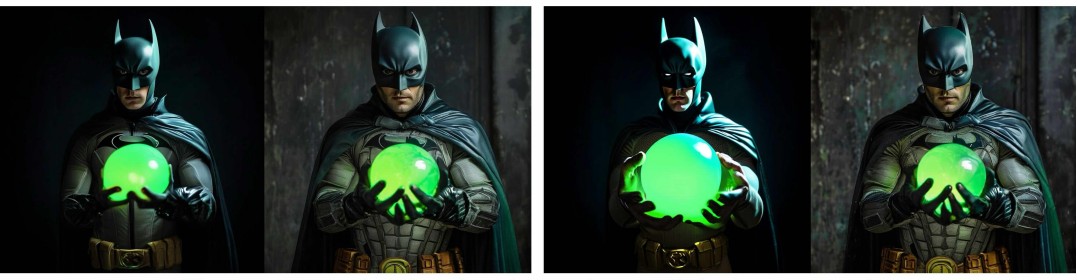

The image is a digital art headshot of an owlfolk character with high detail and dramatic lighting.

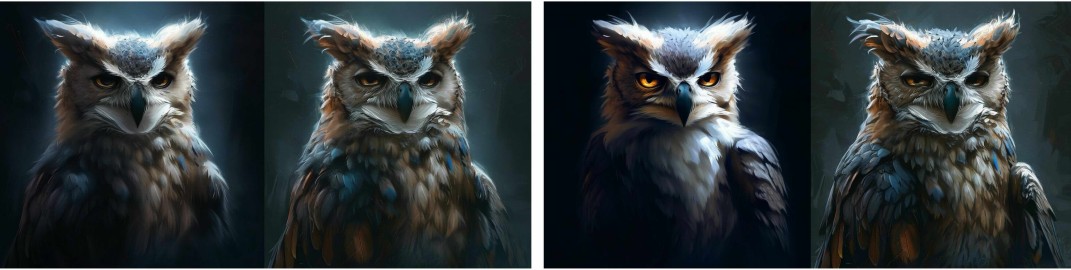

An anthropomorphic cat wearing sunglasses and a leather jacket rides a Harley Davidson in Arizona.

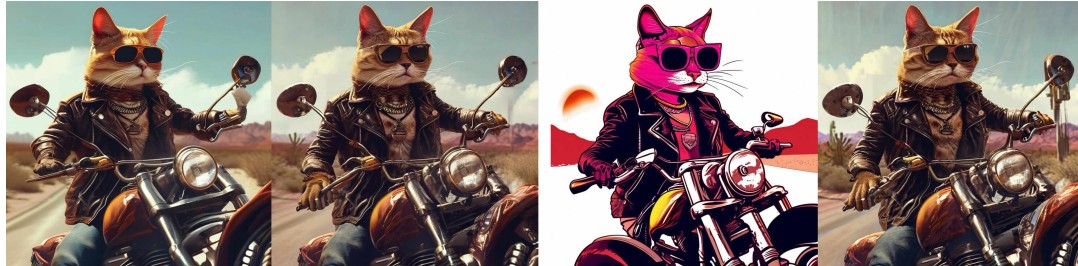

A little girl holding a brown stuffed animal.

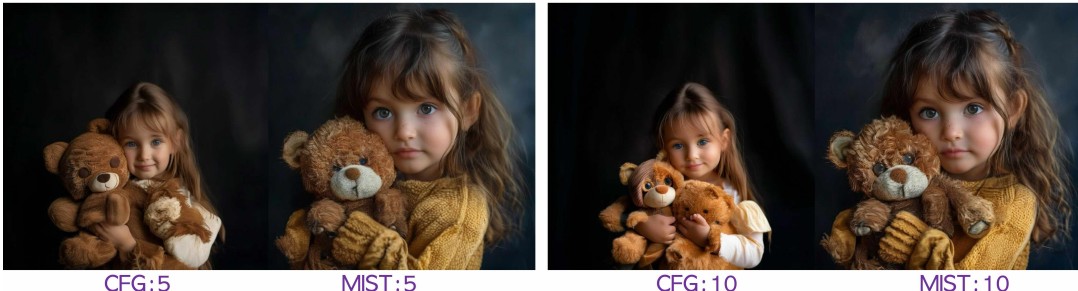

CFG:5  MIST:5  CFG:10  MIST:10

*Figure 12.* Qualitative results on Lumina-Next base model.

