# OpenReview forum: "MIST: Moment-Aligned Invariant Stability Transform for Robust Flow Matching"
_ICML.cc/2026/Conference — ICML 2026 regular_

### Official Review · Reviewer_rTNk · 2026-03-01

**Soundness:** 3
**Presentation:** 3
**Significance:** 3
**Originality:** 3
**Overall Recommendation:** 4
**Confidence:** 4

**Summary:**

MIST addresses the instability of CFG at high guidance scales in flow-matching T2I/T2V models. The paper interprets CFG as an affine perturbation of the learned velocity and shows it induces (i) a linear barycentric drift and (ii) a quadratic energetic instability scaling as w², especially in high-curvature regions. To counter this, MIST introduces a training-free two-stage correction: Invariant Alignment (drift correction + moment matching) and Stability Thresholding (temporal decay + spatial suppression based on a local guidance-to-structure ratio). Experiments across SD3/SD3.5, Lumina/Flux, and Wan2.2 demonstrate consistent improvements over CFG and recent fixes on HPD v2, GenEval, DPG, and VBench, particularly at high guidance.

**Compliance With Llm Reviewing Policy:**

Affirmed.

**Final Justification:**

The rebuttal addressed most of my practical and reproducibility concerns, but since the central theory remains primarily interpretive rather than a formal account of improved sample-distribution fidelity, I maintain my score at 4.

**Key Questions For Authors:**

* How sensitive are IA/ST to solver choice and discretization? For instance, Euler vs Heun, step spacing, and number of steps? A small robustness table would help.
* Given a sample, batch, channel, and spatial dimensions, how are the expectations/variances in IA computed in practice? What is the runtime overhead?
* Were APG / CFG++ / CFG-Zero tuned comparably; i.e., is the same sampling budget, same guidance grid, same prompts/seeds the same? If not, please add a standardized tuning protocol.
* When does MIST hurt? A short failure-mode section would increase confidence.

**Limitations:**

The paper notes incomplete coverage beyond velocity-prediction flow models and that the argument is largely empirical/intuitive. I’d like a clearer discussion of solver/step-count sensitivity and known failure cases.

**Strengths And Weaknesses:**

## Strengths
* The moment-based diagnosis is a useful way to think about why high CFG collapses in flow models.
* MIST is genuinely plug-and-play, and the ablations suggest both IA and ST matter.
* The evaluation is broad, and the gains at higher guidance are often clear.

## Weaknesses
* The theory is mostly an interpretive decomposition; the link from matching velocity moments to improved sample-distribution fidelity is plausible but not really formalized, and the manifold metric term M(x) is more motivational than operational.
* Several baselines have their own tuning knobs and step-schedule dependencies; it’s not always obvious that everything is tuned under the same compute budget and sampling setup.
* Implementation details are a bit under-discussed, for example: (i) how statistics for IA/ST are computed, (ii) any batching costs, (iii) sensitivity to step count or solver.

---

> ### Author Rebuttal · Authors · 2026-03-31
>
> We sincerely thank the reviewer for the careful reading and the emphasis on both theory and reproducibility. We respond point-by-point below.
>
> ### W1: The link to sample-distribution fidelity is not fully formalized, and M(x) is more motivational than operational.
>
> We agree with this assessment. Our theoretical contribution is best viewed as an analysis-motivated decomposition of guidance-induced instability, rather than a full theorem proving that moment matching guarantees improved sample-distribution fidelity. The decomposition is intended to identify two dominant and actionable failure factors, namely mean drift and variance inflation, and to motivate DC and EM accordingly. We also agree that M(x) is currently motivational rather than operational: the implementation uses the Euclidean case, while M(x) provides an information-geometric interpretation of why certain regions may amplify instability. We will revise the text to make this distinction explicit.
>
>
> ### W2&Q3: Baselines have different tuning knobs; fairness is not obvious. Were APG / CFG++ / CFG-Zero tuned comparably?
>
> All comparisons were conducted under a standardized setup for each base model: same base checkpoint, same prompts/seeds, same resolution, same number of sampling steps, same solver, and no extra model evaluations or denoising steps for any method. For all baseline hyperparameters, we followed official implementations or author-recommended settings (as described in D.1 and D.2 in the appendix).
> We will add a concise standardized tuning/fairness paragraph in the revision.
>
>
> ### W3&Q2: Implementation details are under-discussed. How are expectations/variances computed in practice? What is the runtime overhead?
>
> For IA, statistics are computed for each sample at each timestep via reductions over all non-batch dimensions. For ST, the corresponding operations are performed over the channel and spatial dimensions in image models, and additionally over the relevant temporal dimensions in video models.
>
>
> MIST adds no extra network forward passes beyond standard CFG. It only uses lightweight tensor operations, so the overhead is minimal.
> We measured the latency on an H100 GPU.
>
> ||Base|MIST|
> |---|---:|---:|
> |Inference Latency (per step)|325.57 ms|326.32 ms (+0.75 ms)|
>
> MIST adds only 0.75 ms per step, confirming that the practical overhead is negligible.
>
>
> ### Q1: Sensitivity to solver choice and discretization.
>
> Mechanistically, MIST is applied to the predicted velocity before the ODE update, so it is largely solver-agnostic at inference. We have further evaluated MIST with different schedulers, sampling steps, and step spacings, and the gains remain consistent across solver families.
>
> Different solvers:
>
> | Solvers | PickScore | Aesthetic | ImageReward |
> |---|---:|---:|---:|
> | FlowMatchEuler | 22.44 | 5.866 | 1.0361 |
> | FlowMatchEuler + MIST | 23.02 | 6.053 | 1.2216 |
> | FlowMatchHeun | 22.70 | 5.914 | 1.1539 |
> | FlowMatchHeun + MIST | 22.99 | 6.026 | 1.2309 |
> | DPMMultistep | 22.52 | 5.873 | 1.0731 |
> | DPMMultistep + MIST | 23.03 | 6.058 | 1.2388 |
> | UniPCMultistep | 22.26 | 5.810 | 0.9993 |
> | UniPCMultistep + MIST| 22.98 | 6.048 | 1.2284 |
>
> Different sampling steps:
>
> | Steps | PickScore | Aesthetic | ImageReward |
> |---|---:|---:|---:|
> | 20 | 22.14 | 5.773 | 0.8771 |
> | 20 + MIST  | 22.94 | 6.042 | 1.1954 |
> | 28 | 22.44 | 5.866 | 1.0361 |
> | 28 + MIST | 23.02 | 6.053 | 1.2216 |
> | 50 | 22.73 | 5.938 | 1.1720 |
> | 50 + MIST | 23.02 | 6.060 | 1.2369 |
>
> Different time spacing (time shift):
>
> | TimeShift | PickScore | Aesthetic | ImageReward |
> |-----------|-----------|-----------|-------------|
> | 1 | 21.47 | 5.555 | 0.4172 |
> | 1 + MIST| 22.85 | 5.993 | 1.1349 |
> | 3 | 22.44 | 5.866 | 1.0361 |
> | 3 + MIST | 23.02 | 6.053 | 1.2216 |
>
>
> These results suggest that MIST remains beneficial across different sampling settings, including solver types, steps, and time spacings, with consistent gains in all tested settings. We will add a compact robustness table in the revision.
>
>
>
> ### Q2: How are expectations/variances computed in practice? What is the runtime overhead?
> Please see our response to W3 above.
>
>
> ### Q3: Were APG / CFG++ / CFG-Zero tuned comparably?
> Please see our response to W2 above.
>
>
> ### Q4: When does MIST hurt?
>
> We appreciate this suggestion. MIST is most beneficial when the guidance scale is high enough to induce instability. Its gains may be smaller in the following cases:
>
> - very low guidance regimes, where there is little instability to correct
> - cases where stronger local suppression may slightly reduce stylized exaggeration in exchange for improved stability
> - models whose guidance behavior has already been heavily regularized or distilled, leaving less room for further improvement
>
> These cases reflect diminishing returns rather than systematic degradation. We will add corresponding qualitative examples for each of these failure patterns, along with a failure-mode / limitation discussion, in the revision to make these regimes explicit.

---

> > ### Author Rebuttal · Reviewer_rTNk · 2026-03-31
> >
> > Thank you for the detailed rebuttal. The added clarification rebuttal address most of my questions and make the paper stronger. My main remaining reservation is still the one noted in my review: the theory is best viewed as an interpretive decomposition rather than a formal account of improved sample-distribution fidelity. So while the rebuttal improves my confidence and clarifies the intended scope, it does not fully remove that underlying limitation. I found the rebuttal helpful, but I will keep my rating.

---

### Official Review · Reviewer_2rHx · 2026-03-03

**Soundness:** 3
**Presentation:** 2
**Significance:** 2
**Originality:** 2
**Overall Recommendation:** 4
**Confidence:** 4

**Summary:**

This paper proposes a training-free method to stabilize Classifier-Free Guidance (CFG) in flow matching models at high guidance scales. The main challenge is that high guidance scales cause visual artifacts and mode collapse. To address this, the authors propose (1) Invariant Alignment to correct the mean and variance of the guided velocity, and (2) Stability Thresholding to suppress extreme values over time and space. The proposed method is evaluated on T2I benchmarks (GenEval, HPD v2, DPG) and T2V tasks (VBench), showing advantageous performance over CFG variants including APG, CFG++, and CFG-Zero.

**Compliance With Llm Reviewing Policy:**

Affirmed.

**Key Questions For Authors:**

1. How does the proposed method interact with negative prompt or other guidance techniques?
2. Do the two failure modes manifest differently across model architectures, or are they universal to all flow matching models?

**Limitations:**

yes

**Strengths And Weaknesses:**

**Strengths**

1. The proposed method is well motivated, since CFG instability at high guidance scales is an important practical problem that limits flow model usability.
2. The analysis of CFG instability through mean and variance decomposition is novel, identifying two distinct failure modes.
3. Experimental results validate the effectiveness of the proposed method, spanning multiple benchmarks with comparisons against three recent methods and thorough ablations.

**Weaknesses**

1. Lack of motivating experiments. The paper decomposes CFG instability into mean shift and variance change theoretically, but does not verify this on real models (e.g., measuring the two components separately across timesteps). The authors should provide empirical evidence to validate the theoretical claims and motivate the proposed method.
2. Insufficient improvements at moderate guidance scales. On GenEval, the proposed method improves over CFG-Zero by only +0.41 at common guidance scale (5.0), and the advantage becomes meaningful only at extremely high guidance scale at 15.0 (+3.49). Since most scenarios use moderate guidance scales, this casts doubt on the practical significance.
3. Lack of theoretical justification for Stability Thresholding. The temporal decay and spatial suppression are empirically validated but appear ad-hoc. The authors should explain why these design choices are appropriate.
4. Missing computational overhead analysis. The authors describe the proposed method as "training-free" but do not report inference time comparisons. The authors should compare the per-step time cost.

---

> ### Author Rebuttal · Authors · 2026-03-31
>
> We sincerely thank the reviewer for the careful reading and constructive feedback. We address each point below.
>
> ### W1: Lack of motivating experiments.
>
> Thanks for this helpful suggestion.
> We performed timestep-wise diagnostics that track the mean and variance of velocity under different scales, averaged over 500 samples.
>
> Standard CFG (mean/variance):
>
> |Scale\Step|1|5|10|15|20|25|
> |---|---|---|---|---|---|---|
> |CFG 5|0.08/1.72|0.06/1.48|0.06/1.47|0.06/1.47|0.06/1.45|0.06/1.40|
> |CFG 10|0.23/2.63|0.08/1.75|0.08/1.63|0.09/1.61|0.09/1.59|0.08/1.55|
> |CFG 15|0.37/3.65|0.14/2.11|0.14/1.82|0.15/1.77|0.15/1.75|0.15/1.72|
>
> MIST:
>
> |Scale\Step|1|5|10|15|20|25|
> |---|---|---|---|---|---|---|
> |MIST 5|-0.07/1.21|0.02/1.38|0.02/1.40|0.02/1.39|0.02/1.38|0.02/1.31|
> |MIST 10|-0.07/1.21|0.02/1.39|0.02/1.41|0.02/1.41|0.02/1.39|0.02/1.33|
> |MIST 15|-0.07/1.21|0.02/1.39|0.02/1.40|0.02/1.41|0.02/1.39|0.01/1.32|
>
>
> These diagnostics support our claim: as the scale increases, standard CFG exhibits both larger mean shift and stronger variance inflation, whereas MIST keeps both quantities substantially more stable throughout the trajectory.
>
>
> ### W2: Insufficient improvements at moderate guidance scales on GenEval. MIST improves over CFG-Zero by only +0.41 (scale 5), while the advantage is obvious at scale 15.0 (+3.49).
>
> At GS=5, the gain over CFG-Zero is smaller than at GS=15, but it is still non-trivial. MIST improves GenEval from 66.68 to 67.09 over CFG-Zero, i.e., +0.41, while CFG-Zero improves over vanilla CFG only from 66.58 to 66.68, i.e., +0.10. Thus, even at this common moderate scale, MIST provides a larger improvement over CFG-Zero than CFG-Zero provides over vanilla CFG. As a training-free, plug-and-play method, such gains are practically useful.
>
>
> GenEval Overall (Table 7 in paper):
>
> | Method | GS=5 | GS=10 | GS=15 |
> |---|---:|---:|---:|
> | CFG | 66.58 | 67.18 | 59.17 |
> | CFG-Zero | 66.68 | 67.31 | 64.94 |
> | MIST | **67.09** | **68.39** | **68.43** |
>
>
>  MIST’s advantage becomes much clearer at higher guidance scales (GS=15). This matches our design goal, since guidance-induced instability is mild at moderate scales but much more severe at high scales.
>  Thus, MIST not only improves standard settings, but also extends the usable high-guidance regime.
>
>
> ### W3: Lack of theoretical justification for Stability Thresholding (ST). Why these design choices are appropriate.
>
> ST is not an arbitrary heuristic, but a local stability regulator for the residual instability left after global moment alignment.
>
> - **Temporal Decay** is motivated by ODE stability: large step-to-step spikes in guidance can make the dynamics unstable or oscillatory, so damping them enforces practical temporal smoothness.
> - **Spatial Suppression** is motivated by a local relative-perturbation principle: if the guidance becomes too large relative to the unconditional velocity at a location, that region is more likely to leave the stable transport neighborhood and create artifacts.
>
> In this sense, ST complements IA: IA aligns global statistics, while ST limits local dynamical deviations. We will clarify this "global alignment + local stabilization" rationale in the revision and present ST more carefully as an analysis-driven stability prior, rather than a fully derived optimal rule.
>
>
> ### W4: Missing computational overhead (per-step time cost).
>
> MIST adds no extra network forward passes beyond standard CFG. It only uses lightweight tensor operations, so the overhead is minimal.
> We measured the latency on an H100 GPU.
>
> ||Base|MIST|
> |---|---:|---:|
> |Inference Latency (per step)|325.57 ms|326.32 ms (+0.75 ms)|
>
> MIST adds only 0.75 ms per step, confirming that the practical overhead is negligible.
>
>
>
> ### Q1: How does the proposed method interact with negative prompts or other guidance techniques?
>
> MIST is compatible with prompt techniques. We tested two settings:
> 1. adding a negative prompt ("low quality, low res, blurry")
> 2. prompt expansion (PE), where an LLM expands a short prompt into a more detailed one
>
> | Method | PickScore | Aesthetic | ImageReward |
> |---|---:|---:|---:|
> | Base | 22.44 | 5.866 | 1.0361 |
> | Base + Neg Prompt | 22.64 | 5.963 | 1.1494 |
> | MIST + Neg Prompt | 23.00 | 6.102 | 1.2352 |
> | Base + PE | 22.63 | 6.074 | 1.0428 |
> | MIST + PE | 23.20 | 6.274 | 1.2321 |
>
>
> We also combine MIST with CFG-Zero and obtain an ImageReward of 1.2228, comparable to standalone MIST (1.2216). The limited extra gain is expected because both methods correct related guidance instabilities, so the room for additional improvement after one correction is already reduced.
>
>
> ### Q2: Are the failure modes universal to all flow-matching models?
>
> We observe similar failure modes across SD3/3.5, Lumina-Next, and Flux-dev, which is plausible given that they share the same flow-matching sampling paradigm. This is also consistent with our cross-model results (Tables 5,6,10). We will add more qualitative examples in the revision.

---

> > ### Author Rebuttal · Reviewer_2rHx · 2026-04-01
> >
> > The authors addressed my concerns with additional experimental results.

---

### Official Review · Reviewer_HJ8S · 2026-03-11

**Soundness:** 3
**Presentation:** 4
**Significance:** 3
**Originality:** 2
**Overall Recommendation:** 5
**Confidence:** 3

**Summary:**

This paper investigates the instability of Classifier-Free Guidance (CFG) in flow-based generative models at high guidance scales. The authors show that strong guidance induces a distributional shift in the velocity field that can be decomposed into a linear barycentric drift, which shifts the global distribution center, and a quadratic energetic instability, which injects excessive kinetic energy and leads to variance explosion during sampling. To address this issue, the paper proposes MIST (Moment-aligned Invariant Stability Transform), a training-free guidance stabilization method that combines Invariant Alignment (IA) to correct the first and second moments of the velocity field and Stability Thresholding (ST) to locally regulate guidance through temporal decay and spatial suppression. Experiments on multiple text-to-image and text-to-video benchmarks with modern flow-based models demonstrate that MIST improves visual quality and prompt alignment across a wide range of guidance scales and consistently outperforms standard CFG and recent guidance variants.

**Compliance With Llm Reviewing Policy:**

Affirmed.

**Final Justification:**

The authors have addressed the concerns raised in my review during the rebuttal phase, and I have increased my score accordingly.

**Key Questions For Authors:**

1. Some components of MIST, such as variance rescaling in IA and clipping-based regulation in ST, appear conceptually related to normalization or thresholding strategies used in prior diffusion sampling methods. Could the authors clarify how MIST fundamentally differs from these existing approaches? A clearer discussion would help better understand the novelty of the method.

2. The appendix provides ablations for *Tclip* and *γ*, but only under limited configurations. How sensitive is MIST to these hyperparameters across different guidance scales or base models? Providing additional analysis or discussion on the robustness of these parameters would help assess the general applicability of the method.

3.  While Table 1 shows consistent improvements at higher guidance scales, results in other tables (e.g., Tables 4 and 7) indicate that MIST does not always outperform other methods. Could the authors provide further analysis explaining under which conditions MIST is most effective and why its relative performance varies across benchmarks or metrics? Clarifying this would help better interpret the empirical results.

**Limitations:**

yes

**Strengths And Weaknesses:**

### Summary Of Strengths

1. **Clear analytical perspective on CFG instability.**
   The paper provides an intuitive explanation of why large guidance scales lead to instability in flow-based models. This perspective helps explain common artifacts and variance explosions observed during sampling.

2. **Simple and practical method design.**
   The proposed **MIST** method is training-free and easy to integrate into existing sampling pipelines, making it practical for real-world deployment.

3. **Clear presentation and organization.**
   The paper is well structured and easy to follow. The motivation, method description, and experimental results are clearly presented, making the key ideas accessible to readers.

### Summary Of Weaknesses

1. **Potential conceptual overlap with existing stabilization techniques.**
   Some components of the proposed method, such as the variance rescaling in IA and the clipping-based regulation in ST, appear conceptually related to normalization or thresholding strategies used in prior diffusion sampling methods. Although the paper provides a moment-based explanation for CFG instability, a more detailed discussion clarifying how MIST differs from or improves upon existing stabilization techniques would help better highlight the novelty and contribution of the approach.

2. **Limited analysis of hyperparameter sensitivity.**
   The appendix presents ablation studies on several hyperparameters (e.g., *Tclip* and *γ*). However, the analysis remains relatively limited, covering only a small set of parameter configurations and experimental conditions. A more systematic investigation of how these parameters affect performance under different guidance scales or across different base models would help provide a clearer understanding of the robustness of the proposed method.

3. **Inconsistent performance trends across benchmarks and guidance scales.**
   While Table 1 suggests that MIST consistently improves performance as the guidance scale increases, other experimental results (e.g., Tables 4 and 7) indicate that MIST does not always achieve the best results across all metrics or settings. Additional analysis discussing these discrepancies and clarifying under which conditions the proposed method is most effective would help readers better understand its practical applicability.

---

> ### Author Rebuttal · Authors · 2026-03-31
>
> We sincerely thank the reviewer for the thoughtful comments. We respond point-by-point below and will incorporate the relevant clarifications in the revision.
>
>
> ### W1/Q1: Potential conceptual overlap with existing stabilization techniques.
>
> We agree that some components of MIST may resemble normalization or clipping when viewed in isolation.
> However, our contribution is not merely a rescaling/clipping heuristic, but a moment-based view of CFG instability in flow-matching models together with a hierarchical correction design derived from that view.
> Concretely, we attribute instability to (i) first-moment barycentric drift and (ii) second-moment energetic expansion. This motivates IA: Drift Correction removes the mean shift, while Energy Matching restores the variance/energy scale.
> ST then addresses residual local instability via Temporal Decay and Spatial Suppression.
>
> This differs from prior methods in both motivation and mechanism: APG suppresses the parallel CFG component, CFG++ uses a manifold-constrained inverse formulation, and CFG-Zero relies on projection and early-step zero-initialization. In contrast, MIST unifies global statistical alignment and local dynamical stabilization under a single moment-guided framework. We will clarify this distinction in the revision.
>
>
> ### W2/Q2: Limited hyperparameter sensitivity analysis.
>
> We have already included ablations on $T_{clip}$ and $\gamma$ for SD3.5 in the appendix, and we further extend this analysis to multiple models and wider guidance scales.
> In the tables below, each cell is reported in the order SD3/Lumina-Next/Flux-dev.
>
>
>
>
> | $\gamma$ | PickScore | Aesthetic | ImageReward |
> |---|---:|---:|---:|
> | 0.5 | 22.53/22.55/22.94 | 5.953/6.187/6.063 | 1.1621/0.8984/1.2212 |
> | 1.0 | 22.61/22.56/23.02 | 5.967/6.188/6.070 | 1.1659/0.8888/1.2209 |
> | 1.5 | 22.69/22.52/23.03 | 5.988/6.231/6.071 | 1.1462/0.8696/1.2238 |
> | 2.0 | 22.69/22.52/23.11 | 5.966/6.182/6.100 | 1.1378/0.8282/1.2116 |
> | 2.5 | 22.69/22.47/23.10 | 5.961/6.175/6.099 | 1.1065/0.7859/1.2013 |
>
> | $T_{clip}$ | PickScore | Aesthetic | ImageReward |
> |---|---:|---:|---:|
> | 0 | 22.69/22.54/23.08 | 5.988/6.188/6.098 | 1.1462/0.8524/1.2174 |
> | 1 | 22.69/22.52/23.07 | 5.988/6.231/6.085 | 1.1462/0.8696/1.2208 |
> | 2 | 22.65/22.55/23.08 | 5.965/6.180/6.089 | 1.1597/0.8727/1.2269 |
> | 3 | 22.64/22.55/23.03 | 5.962/6.178/6.071 | 1.1661/0.8764/1.2238 |
> | 4 | 22.62/22.55/23.06 | 5.960/6.176/6.081 | 1.1661/0.8764/1.2313 |
>
>
> Due to space limitations, we provide an ablation at guidance scale 15 for SD3 below:
>
> | $\gamma$ | PickScore | Aesthetic | ImageReward |
> |---|---:|---:|---:|
> | 0.5 | 22.18 | 5.855 | 1.1277 |
> | 1.0 | 22.41 | 5.921 | 1.1510 |
> | 1.5 | 22.59 | 5.971 | 1.1582 |
> | 2.0 | 22.62 | 5.945 | 1.1414 |
> | 2.5 | 22.63 | 5.942 | 1.1150 |
>
> | $T_{clip}$ | PickScore | Aesthetic | ImageReward |
> |---|---:|---:|---:|
> | 0 | 22.59 | 5.971 | 1.1583 |
> | 1 | 22.59 | 5.971 | 1.1582 |
> | 2 | 22.52 | 5.939 | 1.1562 |
> | 3 | 22.48 | 5.931 | 1.1563 |
>
>
>
> The key takeaway is that MIST is not brittle: performance changes smoothly rather than sharply as $\gamma$ and $T_{clip}$ vary. Overall, the variation is mild across models and guidance scales, indicating that MIST does not require delicate hyperparameter tuning.
>
>
>
> ### W3/Q3: Inconsistent performance trends across benchmarks and guidance scales. While Table 1 suggests that MIST consistently improves performance, Tables 4 and 7 indicate that MIST does not always achieve the best results across all metrics or settings.
>
> Our claim is not that MIST is best on every single metric or subcategory. Rather, the main takeaway is that MIST delivers consistent overall improvements and stronger robustness across guidance scales, especially in regimes where standard CFG and prior corrections are more prone to instability.
>
> This distinction is particularly relevant for DPG and GenEval. These benchmarks are valuable for measuring fine-grained prompt-following subdimensions (e.g., counting, color attribution, relations, and entity/attribute correctness), but their subcategory scores do not fully reflect the stability and naturalness issues that MIST is designed to address. As a result, a more aggressive guidance strategy may score higher on a narrow semantic subtask while still producing less stable or less natural samples overall.
>
> This is also reflected in the aggregate benchmark scores:
>
> DPG Overall (Table 4 in paper):
>
> | Method | GS=5 | GS=10 | GS=15 |
> |---|---:|---:|---:|
> | CFG | 84.36 | 84.51 | 79.93 |
> | CFG-Zero | 84.97 | 85.29 | 83.40 |
> | MIST | **85.16** | **85.87** | **86.40** |
>
> GenEval Overall (Table 7 in paper):
>
> | Method | GS=5 | GS=10 | GS=15 |
> |---|---:|---:|---:|
> | CFG | 66.58 | 67.18 | 59.17 |
> | CFG-Zero | 66.68 | 67.31 | 64.94 |
> | MIST | **67.09** | **68.39** | **68.43** |
>
> As shown in the tables, MIST achieves clear and consistent improvements in the overall scores across guidance scales, with especially noticeable gains at high guidance.

---

> > ### Author Rebuttal · Reviewer_HJ8S · 2026-04-03
> >
> > The authors have largely addressed my concerns by providing additional experimental results.

---

### Decision · Program_Chairs · 2026-04-30

**Decision:**

Accept (regular)

**Comment:**

This paper received unanimously positive reviews: 5, 4, 4. The reviewers commend the clear presentation, the sensible method design, the novelty, the effectiveness of the approach, and impressively a 'useful way to think about why high CFG collapses in flow models'. There were some minor concerns in the initial draft, but these appear to have been largely addressed in the rebuttal. The AC views this as a clear accept. Congratulations to the authors.